# Semantic Granularity Navigation in Image Editing

**Liangsi Lu** [1]  **Minzhe Guo** [1]  **Xuhang Chen** [2]  **Yang Shi** [1]

## Abstract

Despite the generative capabilities of diffusion and flow models, real-image editing remains constrained by a persistent trade-off between semantic editability and structural fidelity. We trace a primary cause of this limitation to the implicit coupling of edit progress with model scale in existing paradigms. Under this coupling, stronger edits typically require visiting noisier states, which spends computation on destabilizing layout before the semantic change is well localized. We introduce **NaviEdit**, a training-free inference-time controller that decouples edit progress from model scale traversal through a strict self-consistency contract. NaviEdit operates at the rollout level and leaves the underlying pretrained model unchanged. It treats scale as a control input and reallocates a fixed step budget toward semantically responsive intermediate scales instead of destructive high-noise regimes. Experiments show positive average gains across compatible editors and flow backbones, supporting decoupling as a portable inference-time control principle.

## 1. Introduction

Text-to-image (T2I) generators based on diffusion or flow have recently reached a level of realism that makes them attractive as generic visual priors (Lipman et al., 2023; Liu et al., 2022; Peebles & Xie, 2023; Esser et al., 2024). Many image editing methods therefore leverage a pretrained T2I model directly at inference time. SDEdit (Meng et al., 2021) edits by corrupting the source image with noise and denoising it under the generative prior, while Prompt-to-Prompt (Hertz et al., 2023) manipulates cross-attention during denoising to preserve layout while modifying content.

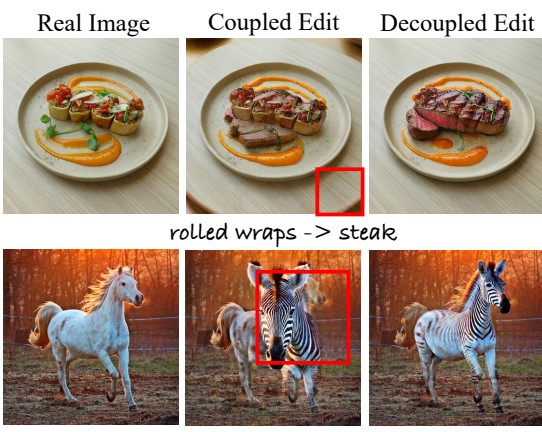

Real Image  Coupled Edit  Decoupled Edit

rolled wraps -> steak

horse -> zebra

*Figure 1.* **Coupling vs. decoupling.** Under the same model and step budget, coupled scale–progress editing exhibits drift/leakage into non-edited regions (red boxes), while decoupled navigation preserves structure while executing the semantic change.

However, a persistent trade-off between editability and fidelity remains: stronger semantic changes often damage non-edited content, whereas stronger preservation often weakens the edit itself. In FlowEdit (Kulikov et al., 2025), for example, moving the trajectory toward higher-noise states affords more semantic freedom but weakens structural anchoring, amplifying artifacts outside the target region (Figure 1). Motivated by this tension, recent methods reshape the sampling process through schedule, resampling, or inversion, including TiNO-Edit (Chen et al., 2024), Schedule Your Edit (Lin et al., 2024) and Dual-Schedule Inversion (Huang et al., 2025).

Many iterative training-free editors make the same implicit design choice: they use the model's *scale coordinate* as an edit clock. The same diffusion timestep, noise level, or flow-time coordinate that indexes the model state is also used as a proxy for how far the edit has progressed. In practice, stronger edits are typically pursued by starting from noisier states or by expanding the visited scale range toward higher noise. Edit progress and scale traversal are therefore coupled by construction.

Our empirical probes suggest that this default assumption is problematic. High scales produce coarse, spatially underdetermined editing, while low scales are dominated by local

[1]Guangdong University of Technology, Guangzhou, China [2]Huizhou University, Huizhou, China. Correspondence to: Yang Shi <sudo.shiyang@gmail.com>; Project page: https://naviedit.github.io.

*Proceedings of the 43rd International Conference on Machine Learning*, Seoul, South Korea. PMLR 306, 2026. Copyright 2026 by the author(s).

reconstruction and struggle to alter geometry. Between them lies an *effective scale window* where the differential field is both semantically responsive and structurally anchored. The scale axis therefore indexes distinct editable-information regimes rather than a uniform progress variable.

We instantiate this view with **NaviEdit**, a training-free controller that treats image editing as controlled navigation of the latent in a differential field under a fixed step budget. A fixed budget edit traverses several such regimes over its trajectory, which we call a *rollout*. We refer to the rollout-level, budgeted property induced by this traversal as *semantic granularity*. NaviEdit navigates these scale-dependent editable-information regimes, and the resulting scale allocation determines the semantic granularity. To do so, NaviEdit introduces an explicit *edit progress axis* and uses the model's scale coordinate as a control input. Conventional pipelines tie progress directly to scale traversal; NaviEdit decouples the two. At each step, NaviEdit selects a scale, queries the co-located differential field at that scale, and advances the latent with a self-consistent update.

Under a fixed step budget, decoupling turns editing into a compute-allocation problem along the scale axis. Rather than expanding the visited range into destructive high-noise regimes, NaviEdit concentrates computation by increasing *density* within the effective scale window; optionally, this density can be adapted online using signals already produced during editing, without additional model evaluations.

Our experiments test this view in three ways: (i) empirical scale diagnostics and effective-window discovery; (ii) controlled schedule comparisons that isolate coupling versus decoupling under a fixed step budget; and (iii) axis-consistency ablations that expose systematic bias under mismatch. We additionally study results across other compatible editors. Finally, we report ablations on compute allocation, optional internal gating, and axis consistency to clarify the conditions required for stable navigation.

## 2. Related Work

Most training-free text-guided editing pipelines built on diffusion-style generators steer sampling along the model's noise/scale coordinate, using noise injection, masked resampling, or attention/feature interventions to trade semantic change against faithfulness; representative examples include SDEdit, RePaint, DiffEdit, Prompt-to-Prompt, Plug-and-Play, and inversion-free editing variants that unify attention control mechanisms (Meng et al., 2021; Lugmayr et al., 2022; Couairon et al., 2022; Hertz et al., 2023; Tumanyan et al., 2023; Xu et al., 2023). Closely related are works that improve reconstruction or edit stability by modifying inversion or noise schedules rather than changing the model, such as Schedule Your Edit and Dual-Schedule Inversion (Lin

et al., 2024; Huang et al., 2025).

On the generative modeling, flow matching and rectified flow view generation as learning vector fields/ODE transport between distributions, and modern text-to-image models increasingly adopt Transformer architectures (e.g., DiT) and rectified-flow Transformers (Lipman et al., 2023; Liu et al., 2022; Peebles & Xie, 2023; Esser et al., 2024). Most recently, FlowEdit demonstrates inversion-free editing directly on pre-trained text-to-image flow models (Kulikov et al., 2025); orthogonally, a line of work shows that frozen diffusion/flow models expose spatially meaningful internal signals that can be mined for localization (e.g., segmentation/correspondence), informing how one might obtain masks without extra training (Luo et al., 2023; Tian et al., 2024).

We provide an extended discussion of related work in Appendix A.

## 3. Empirical Motivation

Current training-free editing pipelines, routinely use the model's scale coordinate as an implicit progress proxy. Stronger semantic edits are pursued by expanding the visited range toward higher noise, while fine details are handled by moving to lower noise. We argue that this conflation of scale (which exposes scale-dependent editable-information regimes) with progress (how much semantic change has been accumulated) is a major contributor to the editability–fidelity tension. Under a fixed model-call budget, editing quality is a property of the entire rollout: it depends on where computation is spent along the scale axis and whether the induced prompt-conditioned motion remains spatially localizable and directionally stable along that traversal. We refer to this rollout-level, budgeted property as semantic granularity, and in Sec. 4 we formalize it as a functional of the trajectory together with its scale allocation.

### 3.1. Probing Spatial Controllability along the Scale Axis

We introduce a diagnostic probe to measure the spatial controllability of the model's semantic prior across scales, independent of any specific editing trajectory. Let $x_{\mathrm{src}}$ denote the source latent, $x$ denote the current editing state, and $u \in [0, 1]$ denote a normalized scale coordinate mapped to the model coordinate via model schedule $t = \tau(u)$ (Appendix C.8). We construct a co-located pair by mixing the source with fresh noise $\epsilon \sim \mathcal{N}(0, I)$ as $z^{\mathrm{src}} = (1 - u)x_{\mathrm{src}} + u\epsilon$ (Lipman et al., 2023; Liu et al., 2022), and aligning the target query by residual shift $z^{\mathrm{tar}} = x + (z^{\mathrm{src}} - x_{\mathrm{src}})$ (Kulikov et al., 2025). Querying the flow model with prompts $(c_{\mathrm{src}}, c_{\mathrm{tar}})$ yields velocities whose difference $\Delta V(u) = v_\theta(z^{\mathrm{tar}}, \tau(u), c_{\mathrm{tar}}) - v_\theta(z^{\mathrm{src}}, \tau(u), c_{\mathrm{src}})$ serves as a local differential field.

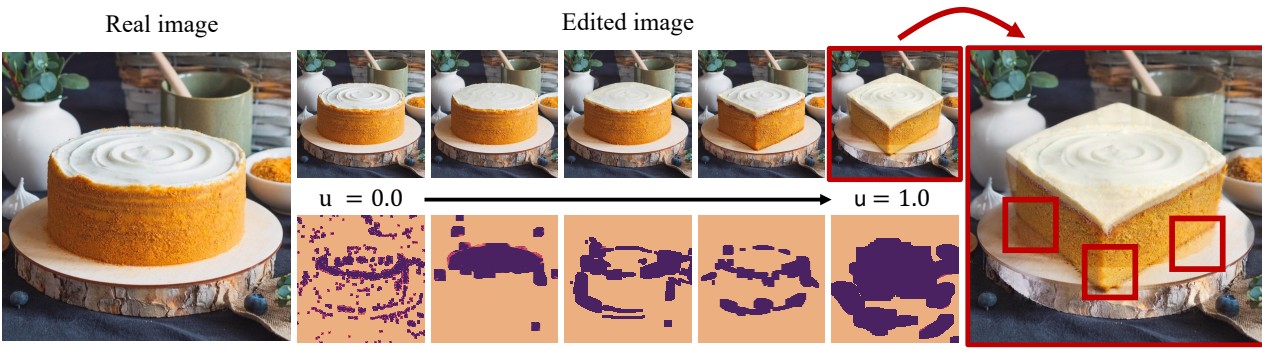

*Figure 2.* **Editable-Information Spectrum and Editability Regimes.** We visualize the differential semantic vector field (bottom) and editing results (top) across scales for the prompt "round cake → square cake". The scale axis reveals a **local editable-information spectrum**: at low scale (small noise scale $u$), the field is **texture-entangled** and fails to alter geometry; at high scale (large noise scale $u$), the field becomes coarse and causes **global drift** with background hallucinations (red boxes). Effective editing requires navigating the **structural sweet spot** (middle), where the field is semantically potent yet spatially anchored.

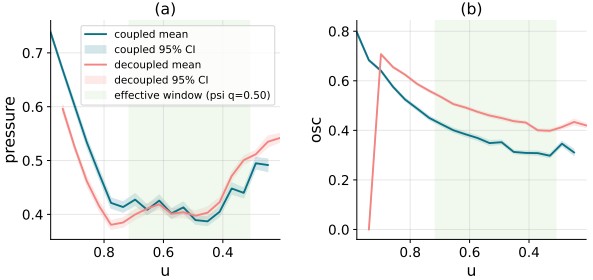

*Figure 3.* **Empirical regime structure on the scale axis.** Sweeping the scale coordinate $u$ and aggregating per-step diagnostics reveals a stable effective window (shaded) where leakage pressure $\rho$ is minimized, flanked by higher-risk tails. (**a**) $\rho$ exhibits a clear valley over $u$. (**b**) The raw oscillation statistic $\omega$ is more schedule-dependent, so we use it as a secondary diagnostic rather than to define the window.

Ideally, $\Delta V(u)$ should concentrate on the target object. A diffuse field indicates that the editing direction is spatially underdetermined, leading to global drift. To quantify this, we use a feasible-region mask $M(u)$ extracted from internal representations, with no external masks and no additional model evaluations beyond the fixed step budget (see Appendix G.1). We summarize the spatial underdetermination by the outside-to-total energy ratio:

$$\rho(u) = \frac{\|(1 - M(u)) \odot \Delta V(u)\|_2}{\|\Delta V(u)\|_2}, \quad (1)$$

where a larger $\rho(u)$ indicates that the prompt-induced motion leaks outside the feasible region, rendering the edit intrinsically hard to localize. Complementarily, we track the local oscillation $\omega(u) = 1 - \cos(\Delta V(u), \Delta V(u - \delta))$ to measure scale-to-scale directional instability for a small $\delta$. Sweeping $u$ while holding the probe construction fixed reveals a robust regime structure on the scale axis. As shown in Figure 3, the leakage pressure $\rho(u)$ exhibits a stable valley at intermediate scales, indicating where prompt-conditioned

motion is most spatially identifiable. We treat this valley as a diagnostic discovery rather than an oracle that must be recomputed at inference. Operationally, NaviEdit restricts navigation to a fixed tail window on the scheduler path, starting from a reference depth (denoted $t_{\mathrm{ref}}$) so as to exclude the extreme high-noise tail. We denote the resulting visited scale interval by $\mathcal{U}_{\mathrm{eff}}$ and report sensitivity in Appendix C.4. In contrast, the oscillation statistic $\omega(u)$ is more schedule-dependent, so we treat it as a secondary diagnostic for discretization and trajectory stability.

### 3.2. The Editable-Information Spectrum and the Coupling Trap

By sweeping the coordinate $u$, we observe that the prompt-conditioned motion induced by the generative prior is spatially non-uniform. In Figure 2, at high scales (large $u$, higher noise), the differential field $\Delta V(u)$ becomes spatially underdetermined: the motion is less confined to plausible edit regions, and leakage pressure $\rho(u)$ increases, correlating with drift and background hallucination. At low scales (small $u$, lower noise), the field is dominated by high-frequency reconstruction constraints, so edits tend to become texture-entangled and struggle to alter geometry; directional behavior can also become brittle, which is captured by increased schedule-dependent oscillations in $\omega(u)$.

Between these extremes lies a semantically responsive region around the valley of $\rho(u)$ in Figure 3. In NaviEdit, we operationalize a fixed visited interval $\mathcal{U}_{\mathrm{eff}}$ by selecting a tail window on the scheduler path (Sec. 3.1), which avoids allocating progress to the extreme high-noise regime under a strict step budget. Within $\mathcal{U}_{\mathrm{eff}}$, the pressure and oscillation diagnostics help explain which scales are more identifiable or more brittle, motivating density allocation under a fixed step budget. We use $\omega(u)$ only to diagnose trajectory sta-

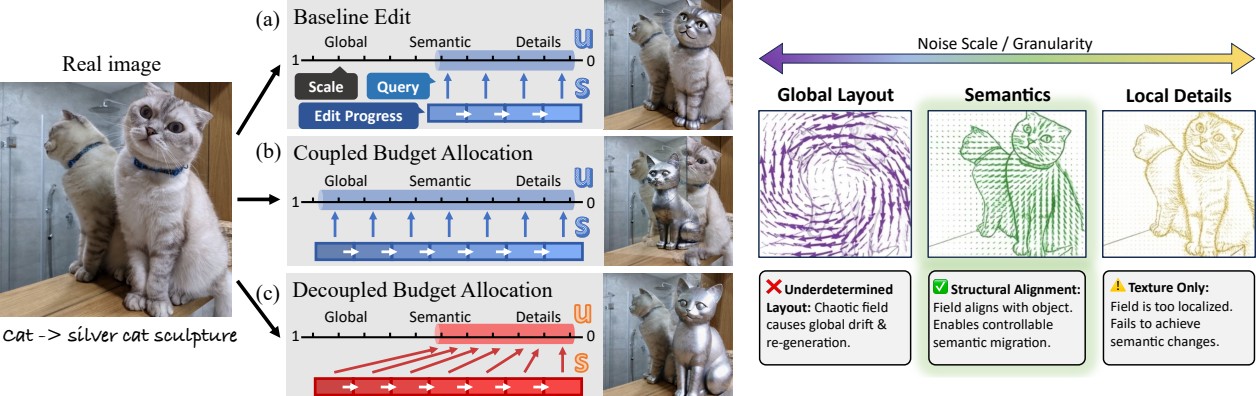

*Figure 4.* Coupling budget with scale range tends to push editing into high-scale regimes where layout is underdetermined. Decoupling budget as density within a fixed effective scale window concentrates computation where the co-located differential field is both strong and spatially identifiable.

bility across scales and to motivate later density-control choices.

This regime structure exposes a limitation of the prevailing design. Conventional schedules often couple edit magnitude with scale-range expansion: to push for stronger edits, they expand the visited range toward the high-noise tail, which allocates nonzero progress mass outside $\mathcal{U}_{\text{eff}}$. In our setting, the high-scale tail is empirically more underdetermined (high $\rho$), and allocating progress mass there correlates with increased drift and artifacts under a fixed model-call budget; this also matches controlled step-budget sweeps where increasing steps under coupled rules improves CLIP but can degrade background fidelity (Appendix D.1). Figure 4 summarizes this coupling trap as a compute-allocation issue rather than an update-rule issue. Recent works on diffusion scheduling (Lin et al., 2024) and flow editing (Kulikov et al., 2025) emphasize the importance of scale allocation, but typically do not separate it from an explicit notion of progress, making the resulting allocation partially dictated by budget-to-range heuristics instead of being a controllable decision.

## 4. Methodological Framework

Section 3 surfaced a recurring paradox under a fixed model-call budget: extending an editing trajectory can reduce structural fidelity and can push the result into a generative failure mode characterized by drift, spurious objects, and tonal explosion. We argue that this is not a tuning artifact but a modeling issue. The diffusion or flow coordinate is a scale axis that indexes what information is malleable, yet many editors treat it as a progress clock that indexes how much change has been accumulated. Once scale is treated as a coordinate $u$ and progress as an explicit system time $s$, editing becomes a controlled integration problem. Under the hard constraints stated in Section 3, the only remaining design

freedom is the allocation of compute along the scale axis.

### 4.1. Editing is a controlled integration problem

We reuse the co-located construction and the differential field from Section 3.1. When a base editor exposes reliable training-free localization signals from the same forward pass, we may optionally form an effective field $\Delta V_{\text{eff}} = \Pi_{M(u)}(\Delta V)$ using an internal gate $M(u)$; otherwise the controller simply uses $\Delta V$ itself. The only conceptual addition is to separate where we measure and actuate, namely the scale coordinate, from how much we actuate, namely progress.

**Definition 4.1** (Progress-Granularity decoupled editing). An editing process is a controlled dynamical system on an explicit progress axis $s \in [0, 1]$:

$$\frac{dx}{ds} = \frac{du}{ds} \Delta V_{\text{eff}}\big(x(s); u(s), \epsilon(s)\big), \quad \epsilon(s) \sim \mathcal{N}(0, I), \quad (2)$$

where $u(s) \in [0, 1]$ is a monotone scale control (noise/scale coordinate).

Definition 4.1 does not assume that an edit can be summarized by a single timestep. An edit is judged by the rollout induced by integrating the co-located differential field under a fixed compute budget.

### 4.2. Rollout semantic granularity is a functional

Section 3.2 shows that reliability is not a property of a single instantaneous direction, but of an entire rollout: the same update rule behaves like controlled semantic migration inside the effective window and like underdetermined re-generation in the high-scale tail. To compare scale allocations under a fixed budget, we evaluate rollouts by a single functional defined operationally for optimization rather than as a theoretical abstraction. It is the quantity that compute

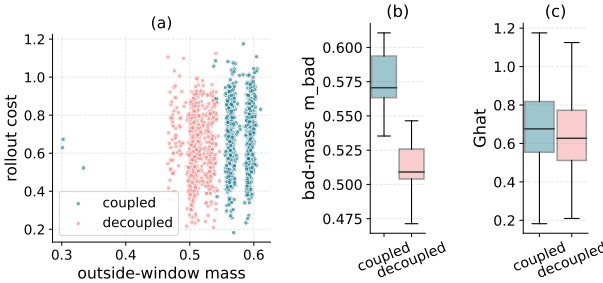

*Figure 5.* **A quantitative illustration of Theorem 4.2.** At a fixed step budget, coupling scale traversal to edit progress forces nonzero outside-window mass $m_{\text{bad}}$, while decoupling concentrates progress within the effective window. **(a)** Rollout proxy cost $\widehat{\mathcal{G}}$ increases with $m_{\text{bad}}$. **(b)** Coupled schedules exhibit larger $m_{\text{bad}}$. **(c)** This forced allocation yields a higher irreducible rollout cost under coupling.

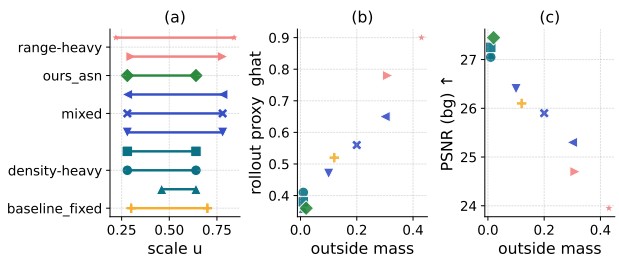

*Figure 6.* **Density vs. range at a fixed step budget.** (a) Scale windows visited by different schedule families. (b) Rollout proxy $\widehat{\mathcal{G}}$ increases with outside-window mass $m_{\text{bad}}$. (c) Background fidelity ($\text{PSNR}_{\text{bg}}$) degrades as $m_{\text{bad}}$ grows, indicating a risk floor induced by range expansion beyond the effective window.

allocation seeks to reduce at a fixed step budget.

We posit a nonnegative local risk density $\phi(x, u)$ with two properties used throughout the paper. First, we assume the local risk density $\phi(x, u)$ increases with the underdetermination measured by our probes in Section 3.1: scales with higher leakage pressure $\rho(u)$ and larger directional oscillation $\omega(u)$ are assigned larger risk. Second, when the visited scales stay inside a fixed effective window, refining density along scale (smaller $\max_k |\Delta u_k|$ at fixed span) does not increase the discretization component of the rollout.

We formalize the semantic granularity of a rollout by the functional $\mathcal{G}[x(\cdot), u(\cdot)] = \int_0^1 \phi(x(s), u(s)) \, ds$. For measurement and online navigation, we instantiate a discrete proxy $\widehat{\mathcal{G}}$ from signals already produced during editing and report its fixed hyperparameters and sensitivity in Appendix C.

### 4.3. Coupling scale and progress is incomplete

Coupling scale traversal to edit progress removes a degree of freedom that becomes necessary once editing quality is judged at the rollout level. Under an identical update rule and identical model-call budget $K$, we compare two sched-

ule families: a coupled family whose budget-to-strength rule expands the visited scale range as edit magnitude increases, and a decoupled family that keeps the visited range within a fixed tail window on the scheduler path and reallocates density within it. Using $\mathcal{U}_{\text{eff}}$ defined by this tail window (Sec. 3.1), we measure the resulting outside-window progress mass $m_{\text{bad}}$ and the rollout proxy cost $\widehat{\mathcal{G}}$. Figure 5 shows a consistent mechanism: $\widehat{\mathcal{G}}$ increases with $m_{\text{bad}}$, coupled schedules exhibit larger $m_{\text{bad}}$, and consequently a higher empirical rollout cost floor at a fixed step budget. This micro evidence matches the measure-allocation argument formalized in Theorem 4.2.

**Theorem 4.2** (Incompleteness induced by coupling). *Assume there exist nonempty intervals $\mathcal{U}_{\text{good}}, \mathcal{U}_{\text{bad}}$ and constants $0 \leq c_{\text{good}} < c_{\text{bad}}$ such that $\phi(x, u) \leq c_{\text{good}}$ on $\mathcal{U}_{\text{good}}$ and $\phi(x, u) \geq c_{\text{bad}}$ on $\mathcal{U}_{\text{bad}}$ for all reachable $x$. Assume every $u(\cdot) \in \mathcal{F}_{\text{couple}}(K)$ assigns at least $\delta(K) > 0$ progress mass to $\mathcal{U}_{\text{bad}}$, with $c_{\text{bad}}\delta(K) > c_{\text{good}}$. Assume also that $\mathcal{F}_{\text{decouple}}(K)$ contains a feasible control supported in $\mathcal{U}_{\text{good}}$. Then, for some editing instances and budgets $K$,*

$$\inf_{u(\cdot) \in \mathcal{F}_{\text{couple}}(K)} \mathcal{G}[x(\cdot), u(\cdot)] > \inf_{u(\cdot) \in \mathcal{F}_{\text{decouple}}(K)} \mathcal{G}[x(\cdot), u(\cdot)]. \tag{3}$$

Coupling forces nonzero progress mass in $\mathcal{U}_{\text{bad}}$, which yields an irreducible contribution to $\mathcal{G}$, while a decoupled control can concentrate allocation within $\mathcal{U}_{\text{good}}$.

### 4.4. Under a fixed step budget, density dominates range

Under fixed compute, the central decision is where to spend model calls along the scale axis. We isolate range from density by fixing the number of evaluations $K$ and the update rule, and varying only the visited scale window. A minimal controlled study shows a consistent directional trend. Expanding the window into high-scale regimes increases drift and artifacts even when edit compliance is matched, whereas refining density within a fixed effective window improves anchoring. Figure 6 visualizes this density versus range effect in a compact setting.

The micro-evidence suggests a structural decomposition. Range expansion inherits a regime-dependent risk floor through nonzero allocation to underdetermined scales, while density refinement within an effective window reduces discretization and active-set instability without paying that floor. We formalize this principle as Theorem 4.3.

Theorem 4.2 implies a concrete compute-allocation principle. When increasing the step budget also expands the visited range into $\mathcal{U}_{\text{bad}}$, the rollout inherits an irreducible risk floor. By contrast, if the visited window stays inside an effective interval, additional steps can be spent on density refinement, reducing discretization error and stabilizing the

active set.

**Theorem 4.3** (Density beats range under a fixed step budget). *Fix an anchor realization and let $\mathcal{G}_K$ be the implemented rollout cost. Assume $\phi \geq 0$, $\phi \leq c_{\text{good}}$ on $\mathcal{U}^\star \subseteq \mathcal{U}_{\text{good}}$, and $\phi \geq c_{\text{bad}}$ on $\mathcal{U}_{\text{bad}}$. For the considered budget $K$, assume every range-expanding rollout has $m_{\text{bad}}^K \geq \delta_K$ with $c_{\text{bad}}\delta_K - c_{\text{good}} \geq \gamma > 0$. Assume also that a self-consistent rollout inside $\mathcal{U}^\star$ has Euler error at most $C_{\text{E}}/K$ and that $\phi$ is $L_\phi$-Lipschitz in $x$ there. If $K > L_\phi C_{\text{E}}/\gamma$, then density refinement inside $\mathcal{U}^\star$ attains lower $\mathcal{G}_K$ than any range-expanding rollout into $\mathcal{U}_{\text{bad}}$.*

Appendix E proves the claim by comparing the implemented bad-regime risk floor with the good-window Euler error bound.

### 4.5. Contract: self-consistent discretization

Theorems 4.2 and 4.3 are statements about allocating scale measure under a fixed compute budget. They only translate to practice if each discrete step measures and actuates at the same scale coordinate. In flow models, the scale coordinate appears in three places within a single step: the noise mixing that constructs the co-located anchors, the model query through the internal timestep mapping $\tau(u)$, and the scale increment used to apply the update. If these operations use inconsistent scale values, the measured differential velocity no longer corresponds to the update that is applied, and the rollout accumulates a systematic bias that manifests as drift and artifacts.

**Theorem 4.4** (Self-consistency contract). *Let $\{u_k\}_{k=0}^{K}$ be a monotone scale sequence and $\epsilon_k \sim \mathcal{N}(0, I)$ be fresh anchors. If a step uses inconsistent scales for (i) mixing, (ii) model querying, and (iii) update increments, then the update does not correspond to any consistent discretization of Definition 4.1 and incurs a systematic bias term that can manifest as drift and artifacts. Conversely, if the step is self-consistent and uses the same $u_k$ for mixing and querying, and uses $\Delta u_k = u_{k+1} - u_k$ for actuation, then the update*

$$x_{k+1} = x_k + \Delta u_k \, \Delta V_{\text{eff}}(x_k; u_k, \epsilon_k)$$

*is a first-order consistent discretization of Definition 4.1 in the effective window.*

The proof is given in Appendix F. We validate Theorem 4.4 with axis-mismatch ablations that independently perturb the scale used for querying, stepping, or mixing while keeping the update rule, budget, and fresh-noise anchoring fixed (see Figure 7).

### 4.6. From theory to method: a rollout-level Navi controller under hard constraints

We instantiate the theory under the constraints in Sec. 3: no training, exactly $K$ model evaluations, and fresh noise

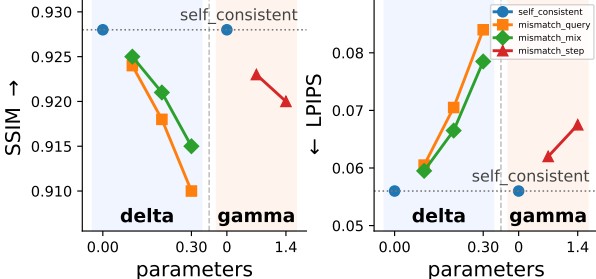

*Figure 7.* Axis-mismatch ablation for Theorem 4.4. Left: qualitative grid as $|\delta|$ increases (mismatch-query uses $u_{k+\delta}$ for model query while keeping mix/update self-consistent). Right: drift and compliance metrics vs $|\delta|$ under fixed $K$.

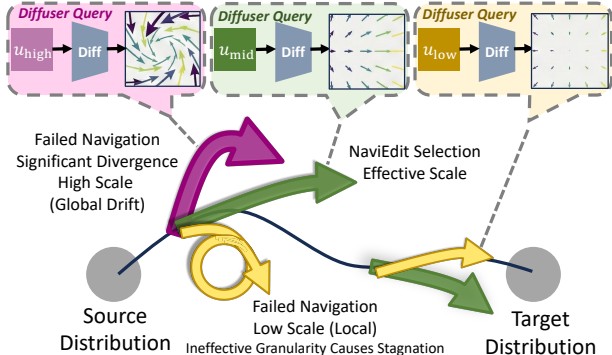

*Figure 8.* **Editing depends on the scale regime.** High scales can diverge while low scales stagnate; NaviEdit allocates density within an effective scale window for stable progress.

at every step. The Navi controller implements the self-consistency contract of Theorem 4.4: each step measures and actuates at the same scale, i.e., co-located mixing, model querying at $t = \tau(u_k)$, and the update step size are all tied to a single $u_k$ and its increment $\Delta u_k$. For naming, we use **NaviEdit** throughout the method development to denote the controller itself, and reserve the shorthand **Navi-X** only for experimental settings where the underlying base editor $X$ must be identified explicitly.

Under this contract, the only remaining design freedom for *progress control* at a fixed step budget is the scale allocation $\{u_k\}$. Guided by Theorem 4.3, we allocate compute by increasing density within an effective scale window instead of expanding range into high-risk regimes. Practically, we parameterize a monotone traversal by a continuous coordinate $p \in [0, 1]$ on the model's discrete path; choosing $\{p_k\}$ induces $\{u_k\}$ and $\{\Delta u_k\}$.

The controller acts only on the rollout: in any compatible prompt-pair editor, it replaces the coupled budget-to-range traversal while leaving the pretrained model unchanged. The running example below uses a FlowEdit/SD3 instantiation, and Appendix H studies additional results on InfEdit and FlowAlign. When an instantiation uses the optional feasible-region gate, we state it explicitly; otherwise we write $M \equiv$

*Table 1.* Quantitative comparison on PIE-Bench. The best/second/third results in each numeric column are highlighted with yellow / orange / blue backgrounds, respectively.

| Type | Method | Struct. Dist.$_{\cdot 10^3}$ ↓ | PSNR↑ | MSE$_{10^3}$ ↓ | SSIM$_{10^2}$ ↑ | LPIPS$_{10^3}$ ↓ | Whole↑ | Edited↑ |
|---|---|---|---|---|---|---|---|---|
| | DiffEdit (Couairon et al., 2022) (SD1.4) | 22.39 | 24.09 | 5.14 | 76.72 | 80.96 | 23.34 | 20.44 |
| | DDIM (Song et al., 2020) + MasaCtrl (Cao et al., 2023) | 28.79 | 21.25 | 8.58 | 80.11 | 106.59 | 24.13 | 21.13 |
| | Direct Inversion (Ju et al., 2023) + MasaCtrl (Cao et al., 2023) | 24.46 | 21.78 | 7.99 | 81.74 | 87.38 | 24.42 | 21.38 |
| Fixed Schedule | DDIM (Song et al., 2020) + PnP (Tumanyan et al., 2023) | 28.20 | 21.26 | 8.42 | 78.90 | 113.58 | 25.45 | 22.54 |
| | Direct Inversion (Ju et al., 2023) + PnP (Tumanyan et al., 2023) | 24.27 | 21.43 | 8.10 | 79.52 | 106.26 | 25.48 | 22.63 |
| | InfEdit (Xu et al., 2023) (SD1.4) | 18.06 | 25.62 | 5.88 | 85.02 | 55.69 | 24.92 | 22.08 |
| | FlowEdit (Kulikov et al., 2025) (SD3) | 14.64 | 22.46 | 7.38 | 84.08 | 103.00 | 25.91 | 22.50 |
| | FlowAlign (Kim et al., 2025) (SD3) | 6.21 | 27.78 | 2.14 | 92.41 | 34.47 | 25.44 | 21.80 |
| | Tino-Edit (Chen et al., 2024) | 17.64 | 22.61 | 6.15 | 82.91 | 87.23 | 25.02 | 22.10 |
| Reschedule | SYE (Lin et al., 2024) (DDIM + PnP) | 27.17 | 21.73 | 8.12 | 87.45 | 110.64 | 24.44 | 21.26 |
| | SYE (Lin et al., 2024) (Direct Inversion + PnP) | 24.11 | 21.57 | 7.94 | 80.24 | 102.62 | 24.52 | 21.39 |
| | TurboEdit (Deutch et al., 2024) (SDXL-Turbo) | 13.80 | 21.44 | 9.49 | 80.08 | 108.60 | 24.66 | 21.70 |
| | **Navi-FlowEdit ($M \equiv 1$) (Kulikov et al., 2025) (SD3)** | 14.25 | 22.54 | 6.86 | 89.36 | 92.47 | 26.01 | 22.59 |
| Navi Controller | **Navi-FlowEdit + gate (Kulikov et al., 2025) (SD3)** | 10.67 | 27.94 | 2.64 | 93.85 | 48.74 | 26.18 | 22.72 |
| | **Navi-FlowAlign ($M \equiv 1$) (Kim et al., 2025) (SD3)** | 5.40 | 28.33 | 2.09 | 93.40 | 34.49 | 26.15 | 22.44 |

1. In this paper, the default FlowEdit system is reported as **Navi-FlowEdit + gate**, while **Navi-FlowEdit ($M \equiv 1$)**, **Navi-InfEdit ($M \equiv 1$)**, and **Navi-FlowAlign ($M \equiv 1$)** denote ungated variants.

Optionally, we adapt the next step along $p$ online using only already-computed per-step signals (a discrete proxy of $\phi$), without additional model calls. The complete inference pipeline is given in Appendix J (Algorithm 1).

# 5. Experimental Validation

## 5.1. Experimental Setup

**Dataset and evaluation metrics.** We evaluate on two complementary benchmarks. PIE-Bench (Ju et al., 2023) serves as the main preservation benchmark: it comprises 700 images with ground-truth masks. We assess performance using Structure Distance for global geometry, PSNR, MSE, SSIM, and LPIPS for background fidelity (computed on non-edited regions), together with CLIP-Whole/Edited scores for semantic alignment. To test the controller beyond masked local editing, we also evaluate on ImgEdit-Bench (Ye et al., 2025), following its Basic and UGE protocols.

**ImgEdit-Bench adaptation.** ImgEdit-Bench provides a source image and an edit instruction, whereas prompt-pair differential editors require both a source description and a target description. We therefore use a single processed-annotation layer generated with Qwen2.5-VL (Bai et al., 2025), which converts each sample into shared (src_prompt, tar_prompt) pairs. InfEdit, FlowAlign, and their Navi variants all consume this same adaptation. Following ImgEdit-Judge, we report per-category Basic scores together with the Basic and UGE averages; the multi-turn subsets are retained only for qualitative inspection.

**Naming convention.** In the main text, *NaviEdit* denotes the controller/framework. We use labels such as *Navi-FlowEdit* and *Navi-FlowAlign* only when a table, benchmark, or cross-editor experiment needs to distinguish which base editor the controller is instantiated on. When the optional feasible-region gate is enabled, we write it explicitly as + *gate*; when it is disabled, we write $M \equiv 1$.

**Implementation details.** All experiments are conducted on a single NVIDIA RTX 3090 GPU. The controller is entirely training-free. It decouples the editing progress axis from the model scale coordinate and navigates a fixed tail window on the scheduler path, denoted by $\mathcal{U}_{\text{eff}}$ and parameterized by $t_{\text{ref}}$ (Appendix C.3). In the main PIE-Bench comparison, the three Navi-controller rows in Table 1 use schedule step $N = 50$, editing budget $K = 50$, and $t_{\text{ref}} = 42$. The ImgEdit-Bench transfer results use ungated *Navi-InfEdit ($M \equiv 1$)* and *Navi-FlowAlign ($M \equiv 1$)*. For fair comparison, Table 1 uses each baseline's official configuration and recommended backbone, e.g., SD1.4 for DiffEdit/InfEdit, SDXL-Turbo for TurboEdit, SD3 for FlowEdit, and SD3 for FlowAlign. Baselines use their official editing-step settings. PIE-Bench provides ground-truth masks; we use them only to define edited and background pixels for evaluation, and do not feed any external masks into the Navi controller or any baseline.

## 5.2. Comparison with Prior Methods

**Quantitative Results on PIE-Bench.** Table 1 reports the strongest deployed systems together with one controlled decomposition pair. The default FlowEdit instantiation, *Navi-FlowEdit + gate*, substantially improves over official FlowEdit on every reported PIE-Bench metric. The two FlowEdit-based Navi rows, *Navi-FlowEdit ($M \equiv 1$)* and *Navi-FlowEdit + gate*, isolate the role of the optional gate: under the same controlled setting, the gate mainly improves preservation. The ungated FlowAlign row, *Navi-FlowAlign ($M \equiv 1$)*, shows that the same decoupled controller re-

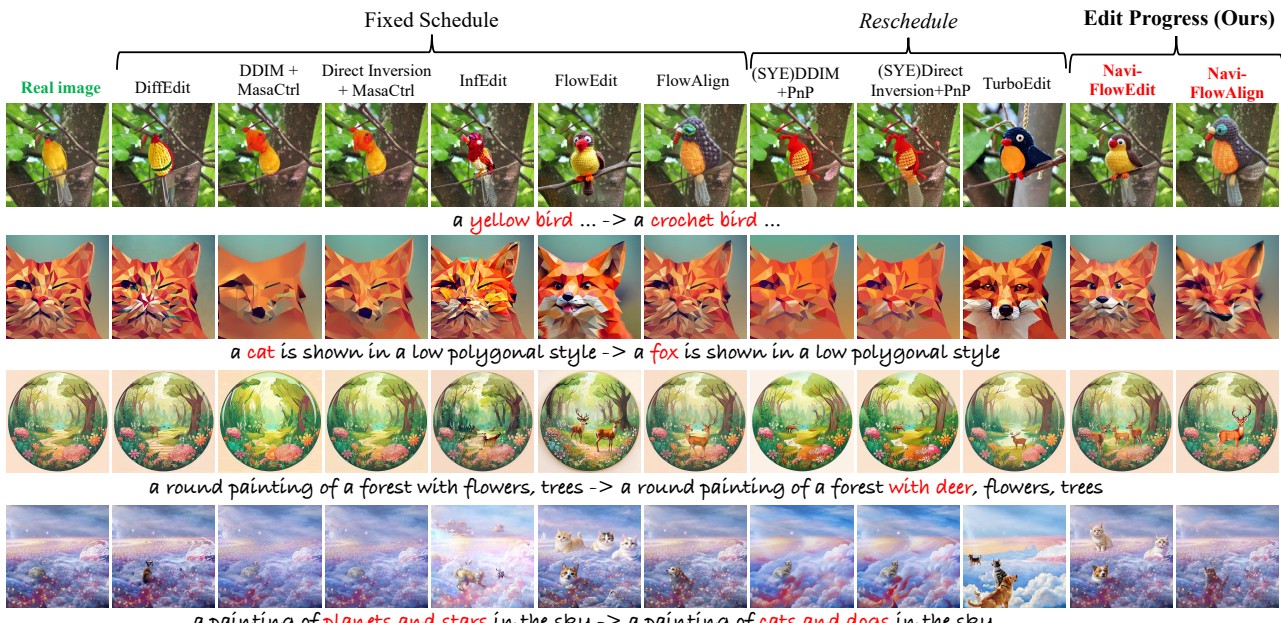

*Figure 9.* **Comparison of edited results.** Real images are in the first column. Prompts are noted under each row.

mains effective even without any additional gating module. Among the deployed rows, *Navi-FlowAlign (M ≡ 1)* attains the best Dist/PSNR/MSE and competitive SSIM/LPIPS, while *Navi-FlowEdit + gate* attains the best SSIM and the highest CLIP-Whole/CLIP-Edited scores.

**ImgEdit-Bench Results Across Compatible Editors.** Figure 10 summarizes category-wise results on ImgEdit-Bench across compatible editors. Both Navi variants are ungated, so the gains isolate the decoupled controller rather than the optional gate. Navi improves both the Basic and UGE averages for InfEdit and FlowAlign. The gains are broad for InfEdit. For FlowAlign, the average gains are smaller and not every Basic category improves, but the strongest positive changes appear in background, action, and replace, where trajectory drift is more likely to matter. This supports our claim that the controller remains effective across compatible editors rather than only in the original FlowEdit instantiation. Full category-wise scores are reported in Appendix H.

**Qualitative Results on PIE-Bench.** Figure 9 visualizes typical outcomes on PIE-Bench. *Navi-FlowEdit + gate* realizes large semantic changes while keeping non-edited regions stable, whereas coupled baselines more often exhibit global drift or background artifacts. Appendix D.1 isolates the effect of decoupled scale allocation under a fixed gating configuration, while Table 1 and Appendix G.4 show that the optional gate mainly improves background preservation. Appendix H provides the full ImgEdit-Bench breakdown, and the user study (Appendix L) is consistent with these trends.

*Table 2.* **Across flow backbones, controlled** $K = 28$ **ablation.** Using the same differential editing mechanism and matched $K = 28$ budget, we compare coupled and decoupled scale allocation on SD3, SD3.5, and FLUX.1 [dev]. Decoupling improves SSIM/PSNR across all three backbones while keeping CLIP-Whole/Edited comparable or slightly higher.

| Method | Background | | CLIP Semantics | |
|---|---|---|---|---|
| | SSIM ↑ | PSNR ↑ | Whole ↑ | Edited ↑ |
| SD3 (Esser et al., 2024) (couple) | 88.22 | 22.18 | 26.01 | 22.55 |
| SD3 (Esser et al., 2024) (decouple) | **93.22** | **27.81** | 26.15 | 22.67 |
| SD3.5 (Esser et al., 2024) (couple) | 85.68 | 22.01 | 26.57 | 22.91 |
| SD3.5 (Esser et al., 2024) (decouple) | 92.32 | 27.45 | 26.77 | 23.32 |
| FLUX.1 [dev] (Labs, 2024) (couple) | 82.14 | 21.81 | 27.02 | 23.35 |
| FLUX.1 [dev] (Labs, 2024) (decouple) | 91.75 | 26.83 | **27.06** | **23.42** |

## 6. Discussions

**Can rescheduling replicate NaviEdit?** No. The distinction is not the shape of the schedule, but the explicit separation of progress from scale together with the self-consistent step contract. Once progress is modeled as its own axis, the design problem becomes scale allocation under a fixed budget. Any rule that mixes, queries, and updates at inconsistent scales is not a valid discretization of the controlled system and empirically leads to the drift and artifacts seen in coupled baselines.

**Is NaviEdit merely tuning? Frontier shift rather than a single best point.** Appendix D.1 (Figure 14) shows a *family-level* shift: at comparable CLIP-Whole, decoupled schedules attain higher background fidelity than coupled ones, suggesting a cost floor induced by forced exposure to risky regimes under coupling. This is exactly the empirical

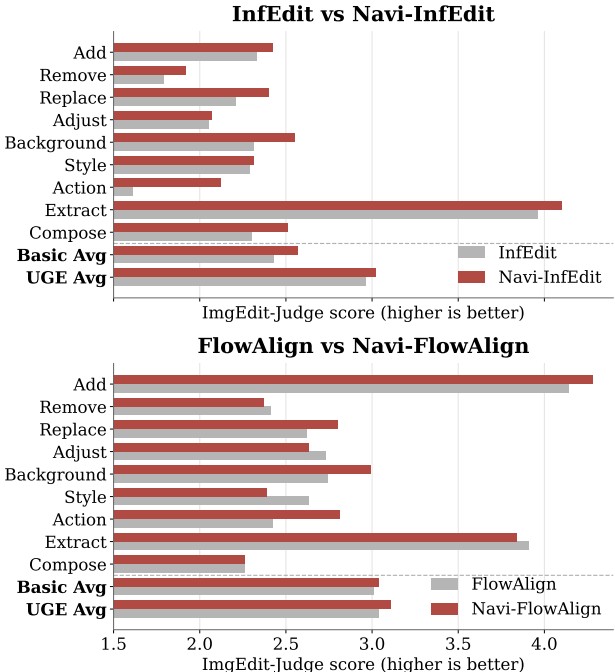

**InfEdit vs Navi-InfEdit**

**FlowAlign vs Navi-FlowAlign**

*Figure 10.* **Category-wise results on ImgEdit-Bench across compatible editors.** Top: InfEdit vs. Navi-InfEdit. Bottom: FlowAlign vs. Navi-FlowAlign. Both Navi variants use $M \equiv 1$. The dashed separator isolates the Basic and UGE averages from the nine Basic categories.

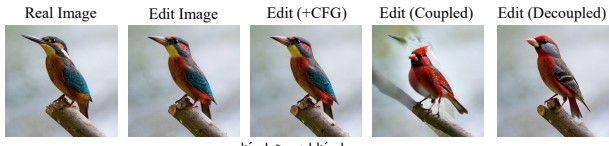

*Figure 11.* **Limits of increasing classifier-free guidance (CFG).** Higher CFG does not reliably prevent drift under coupling, while decoupling preserves structure at similar edit strength.

signature predicted by a rollout-level view: under a fixed step budget, performance is governed by where progress mass is spent along scale, not by any single timestep choice.

**Across flow architectures.** The Navi controller assumes only a conditional velocity field and a monotone scale path. It does not depend on a particular backbone. Table 2 provides a controlled cross-backbone ablation for the same editing mechanism across SD3, SD3.5, and FLUX.1 [dev]. In all three cases, decoupling improves SSIM/PSNR over the coupled allocation and keeps CLIP-Whole/Edited comparable or slightly higher. The edit-scale mismatch is therefore not an idiosyncratic artifact of one model family.

**Why not just increase CFG?** Increasing CFG strengthens conditional alignment, but it does not address the progress–scale coupling: it changes the *magnitude* of the driving field, not the *allocation* of compute along scale under a fixed

step budget (Ho & Salimans, 2022). Figure 11 illustrates the practical implication in editing: simply raising CFG does not reliably reproduce the decoupled improvement, while coupled schedules may still drift/regenerate; decoupling achieves the semantic change with substantially better anchoring at comparable guidance.

## 7. Limitations

NaviEdit improves *how* computation is allocated along the editing rollout, but it does not by itself solve support estimation or scene reasoning. When the editable support is too conservative, or when a fixed effective window is too preservation-biased for a difficult replacement, the result can remain semantically intermediate; this is most visible in fine-grained replacements and can be accentuated by gated variants. NaviEdit does not impose explicit geometric or relational consistency, so scenes involving mirrors, reflections, repeated objects, or strong inter-object relations may remain globally inconsistent even when local drift is suppressed. More broadly, the controller is most effective when the editability-preservation tradeoff is still strongly governed by allocation; for strong trained image editors that already learn part of this tradeoff end-to-end, the room for additional gains can be smaller. Finally, the current controller is defined for compatible prompt-pair editors that expose a conditional differential field and a monotone scale path; extending the same control principle to less aligned editing interfaces remains future work.

## 8. Conclusion

We revisited training-free, inversion-free real-image editing with drift-based generative editors through a single mismatch: existing pipelines implicitly use the model scale coordinate as both a granularity coordinate and an edit-progress clock. By separating progress from scale and enforcing a strict self-consistency contract, NaviEdit turns editing into a rollout-level vector-field navigation controller under a fixed step budget. This makes NaviEdit a plug-and-play inference-time controller for compatible drift-based editors: it does not require training, inversion, masks, or modifying the underlying generative model. The resulting allocation principle is simple: spend compute densely within an effective scale window rather than expanding into high-noise regimes. Empirically, the FlowEdit instantiation *Navi-FlowEdit + gate* substantially improves over its coupled counterpart, while ungated transfer results with *Navi-InfEdit ($M \equiv 1$)* and *Navi-FlowAlign ($M \equiv 1$)* show positive average gains beyond the original FlowEdit setting. These results suggest that the key contribution is not a specific heuristic, but a general progress-scale decoupling principle for training-free drift-based editing.

## Impact Statement

This paper presents work whose goal is to advance the field of Machine Learning, specifically in controllable image editing. The proposed method may support creative and assistive applications by improving editing controllability and reducing the need for task-specific training. As with other image editing and generative modeling techniques, the method should be used responsibly, since edited images may be misinterpreted or misused in inappropriate contexts. We discuss broader societal considerations in Appendix K.

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

# A. Full Related Work

Training-free image editing with diffusion or flow priors is often framed as intervening on a pretrained generative trajectory while preserving a user-provided source image. A recurring practical pattern is to use the model's native coordinate (diffusion timestep, noise level, or flow time) both as a semantic-scale knob and as an implicit notion of edit progress, by deciding where to inject structure constraints and where to apply semantic change. Our work is aligned with this line in that we remain training-free and operate by manipulating model-implied vector fields, but we differ in the organizing principle: we treat the model coordinate as a scale (granularity) coordinate and introduce an explicit edit-progress axis that is independent of scale allocation.

SDEdit injects noise into a source image and then runs the reverse stochastic process to obtain a realistic sample that remains faithful to the corrupted source, using the noise magnitude as the main knob to trade off faithfulness versus realism (Meng et al., 2021). RePaint adapts reverse diffusion iterations for inpainting by repeatedly resampling unmasked regions from the source while denoising the masked region, again using the diffusion trajectory and its step allocation as the operational model (Lugmayr et al., 2022). These works highlight that trajectory intervention and step allocation can be sufficient for strong editing behavior without additional training, but they also exemplify that the trajectory coordinate simultaneously mediates both which information is preserved and how much semantic change is allowed.

A complementary family of training-free editors manipulates internal representations, especially attention and intermediate features, to preserve structure while applying text-driven changes. Prompt-to-Prompt observes that cross-attention mediates the association between words and spatial layout, and edits are realized by controlling cross-attention maps along the generation process; it also introduces localized control mechanisms such as LocalBlend based on attention maps (Hertz et al., 2023). Plug-and-Play Diffusion Features injects spatial features and self-attention information extracted from a source image into the target generation to retain layout and structure (Tumanyan et al., 2023). DiffEdit constructs an edit mask by contrasting diffusion predictions under different prompts and then performs masked semantic editing (Couairon et al., 2022). In instruction-guided editing, ZONE leverages localization cues in instruction-conditioned diffusion models to enable zero-shot local edits (Li et al., 2024). These methods provide strong empirical evidence that internal attention/feature channels encode controllable localization signals and that editing can be staged by choosing where in the model's trajectory these signals are injected.

Several recent works explicitly target the fidelity–editability tradeoff by adding constraints or schedulers that decide at which timesteps to emphasize preservation versus semantic change. UnifyEdit formulates training-free editing as diffusion latent optimization with a self-attention preservation constraint and a cross-attention alignment constraint, and proposes an adaptive timestep scheduler to balance the two in a unified mechanism (Mao et al., 2025). Our focus is orthogonal: rather than treating timestep selection as the only axis for balancing objectives, we explicitly separate edit progress from the scale coordinate and study budget allocation along the scale axis as a controllability problem under a fixed computational budget.

Flow-based and rectified-flow models have recently become central to high-capacity text-to-image systems, and a number of training-free editors are being rederived for this setting. Stable Flow studies training-free editing on DiT-style flow-matching models and identifies vital layers for selective attention injection, addressing the practical question of where to intervene in transformer-based flow models (Avrahami et al., 2025). FlowEdit proposes an inversion-free and optimization-free approach that constructs an ODE mapping between the source- and target-prompt distributions for pretrained flow models, motivated by reduced transport cost relative to edit-by-inversion (Kulikov et al., 2025). RF-Solver and RF-Edit improve numerical solution accuracy of the rectified-flow ODE and build inversion/editing pipelines atop the improved solver (Wang et al., 2024). These works are closely related in spirit in that they treat editing as trajectory manipulation for rectified flows; our contribution is a different abstraction layer, emphasizing explicit separation between an edit-progress parameter and the model's scale coordinate, and making scale-density allocation an object of analysis rather than a byproduct of step count and default schedules.

Beyond trajectory-level interventions, some approaches decompose either the edit itself or the latent degrees of freedom into more structured components. Adjacent to image editing, composed image retrieval also studies how textual modification instructions interact with visual content, including explicit parsing of fine-grained modification semantics (Li et al., 2025b). IEAP decomposes instruction-driven editing into a sequence of atomic operations implemented via lightweight adapters and orchestrated as an editing program, enabling compositional multi-step edits under a common DiT model (Hu et al., 2025). This addresses a different axis of complexity—semantic compositionality over operations—while our method focuses on how to allocate a fixed computational budget across scales for a given operation. Kouzelis et al. propose unsupervised local manipulation by analyzing Jacobians of a pretrained diffusion model to identify local latent directions via joint/individual

component analysis (Kouzelis et al., 2024). HybridEditDif introduces a dynamic decoupled cross-attention mechanism to combine text and exemplar conditioning for guided editing, targeting consistency and controllability via architectural conditioning design (Liu et al., 2025). These works suggest that disentangling and localizing degrees of freedom—via program structure, latent directions, or decoupled conditioning—can substantially improve controllability, and they motivate our emphasis on separating the control over scale (granularity) from the notion of edit progress.

Our appendix also relies on the broader literature showing that diffusion and flow models contain rich internal representations that support localization and diagnostic signals. DIFT extracts diffusion features as correspondence descriptors without task-specific training, showing that diffusion networks encode semantically meaningful spatial structure (Tang et al., 2023). Diffusion Hyperfeatures aggregate multi-timestep and multi-layer diffusion features into per-pixel descriptors for semantic correspondence, formalizing that information is distributed across both space and the diffusion coordinate (Luo et al., 2023). DiffSeg segments images in a training-free, zero-shot manner by merging self-attention maps in Stable Diffusion (Tian et al., 2024). MaskDiffusion revisits cross-attention and proposes training-free masking to improve text-image consistency, explicitly linking attention-map pathologies to semantic mismatch (Zhou et al., 2025). In our framework, attention-derived masks and pressure-like signals are treated as diagnostics for identifying effective scale windows and feasible edit regions, rather than as additional learnable modules.

Finally, diffusion bridges and Schrödinger-bridge formulations provide a conceptual lens where conditional generation is viewed as transporting between endpoint distributions. I2SB introduces an image-to-image Schrödinger bridge formulation for learning diffusion processes between two distributions (Liu et al., 2023), and subsequent works develop implicit or restoration-oriented SB formulations (Wang et al., 2025). While these methods are typically training-based and focus on learning the bridge dynamics, they support the general viewpoint that editing can be framed as controlled transport; our method operationalizes a training-free version of this view by directly manipulating pretrained flow vector fields with explicit separation between scale control and edit progress.

Although these adjacent directions are outside our direct experimental scope, they reinforce the broader methodological pattern that controllable generation benefits from explicit intermediate structure rather than end-to-end output correction alone. In vector graphics and animation, rendering-aware self-feedback and sparse state modeling similarly aim to preserve geometric or temporal structure while changing high-level content (Liang et al., 2026a;b). In other structured generative or adaptation settings, structure–semantic evolution trajectories and geometry-aware diffusion recommenders show that controllability often depends on matching the representation geometry to the task domain (Chen et al., 2026a; Yuan et al., 2025). Related motivations also appear in scientific reconstruction and multimodal reasoning, where agentic tool use, representation engineering, context-aware attention modulation, and confidence-based relational alignment constrain outputs through intermediate signals rather than only through final-sample supervision (Yang et al., 2025; Li et al., 2025a; 2026; Chen et al., 2026b).

## B. Symbols Table

*Table 3.* Symbols Table

| Symbol | Description |
| --- | --- |
| *Axes, schedules, and budgets* | |
| $K$ | Step budget; for NaviEdit this equals the number of model evaluations. |
| $N$ | Number of scheduler timesteps. |
| $k$ | Edit-step index ($k = 0, \ldots, K - 1$). |
| $n$ | Scheduler index ($n = 0, \ldots, N - 1$). |
| $s \in [0, 1]$ | Explicit progress axis (ODE time). |
| $u \in [0, 1]$ | Normalized scale / noise coordinate (granularity axis). |
| $t$ | Model timestep coordinate. |
| $\tau(u)$ | Timestep map (used as $\tau(u) = uT$). |
| $T$ | Scheduler training-time timestep scale. |
| $\{t^{(n)}\}_{n=0}^{N-1}$ | Scheduler timesteps at inference. |
| $u^{(n)}$ | Normalized scale, $u^{(n)} = t^{(n)}/T$. |
| $(t_{\mathrm{path}}[n], u_{\mathrm{path}}[n])$ | Monotone scheduler path (noisy→clean). |

Table 3 – continued from previous page

| Symbol | Description |
| --- | --- |
| $p \in [0, 1]$ | Navigation coordinate along the scheduler path (decoupled from $u$). |
| $p_k$ | Navigation coordinate at step $k$. |
| $\Delta p_k$ | Increment of $p$ at step $k$. |
| $\Delta p_{\min}$ | Base remaining-progress increment, $(1 - p_k)/(K - k)$. |
| $\epsilon_p$ | Small endpoint clamp used before the terminal step. |
| $\text{Interp}(\cdot\,; p)$ | Linear interpolation from $p$ to $(t, u)$. |
| $\text{Speed}(\text{risk})$ | Online speed controller (risk$\rightarrow$step density). |
| risk | Online navigation risk signal (from existing diagnostics). |
| $t_{\text{ref}}$ | Reference depth for the tail window. |
| $n_0$ | Tail start index, $n_0 = \max(0, N - t_{\text{ref}})$. |
| *Latents, anchors, conditions* | |
| $x_{\text{src}}$ | Source latent. |
| $x$ | Current latent state. |
| $x_k$ | Latent at step $k$. |
| $x_K$ | Final edited latent. |
| $x(s)$ | Continuous latent trajectory. |
| $c_{\text{src}}$ | Source condition / prompt. |
| $c_{\text{tar}}$ | Target condition / prompt. |
| $\epsilon \sim \mathcal{N}(0, I)$ | Gaussian noise anchor. |
| $\epsilon_k$ | Fresh noise at step $k$. |
| $I$ | Identity matrix. |
| $z^{\text{src}}(x_{\text{src}}; u, \epsilon)$ | Co-located source anchor, $(1 - u)x_{\text{src}} + u\epsilon$. |
| $z^{\text{tar}}(x; x_{\text{src}}; u, \epsilon)$ | Target anchor via residual shift. |
| $z_k^{\text{src}}$ | Step-$k$ source anchor. |
| $z_k^{\text{tar}}$ | Step-$k$ target anchor. |
| *model field and differential fields* | |
| $v_\theta(\cdot)$ | model velocity field. |
| $\theta$ | model parameters. |
| $V_k^{\text{src}}$ | Source velocity at step $k$. |
| $V_k^{\text{tar}}$ | Target velocity at step $k$. |
| $\Delta V(u)$ | Local differential field at scale $u$. |
| $\Delta V_k$ | Step-$k$ differential velocity, $V_k^{\text{tar}} - V_k^{\text{src}}$. |
| $M(u)$ | Feasible-region mask at scale $u$. |
| $M_k$ | Step-$k$ mask ($\in [0, 1]^{H \times W \times C}$). |
| $\Pi_{M(u)}(\cdot)$ | Active-set projection induced by $M(u)$. |
| $\Delta V_{\text{eff}}(x; u, \epsilon)$ | Effective field (projected differential field). |
| $\Delta V_k^{\text{eff}}$ | Masked field, $\Delta V_k^{\text{eff}} := M_k \odot \Delta V_k$. |
| $\odot$ | Elementwise (Hadamard) product. |
| *Discretization / self-consistency* | |
| $u_k$ | Scale used at step $k$ (mixing/query/update share it). |
| $t_k$ | Timestep queried at step $k$ (from interpolation / $\tau(u_k)$). |
| $\Delta u_k$ | Scale increment, $\Delta u_k = u_{k+1} - u_k$. |
| $\Delta s_k$ | Progress weight from scale steps. |
| $\tilde{u}_k$ | Mismatched query scale (used in contract violation). |
| $\widetilde{\Delta V}_{\text{eff}}(x; u, \tilde{u}, \epsilon)$ | Mismatch-field (anchors at $u$, query at $\tilde{u}$). |
| $b_k$ | Bias term induced by mismatch. |
| $x_{k+1}^{\text{sc}}$ | Self-consistent Euler update result. |
| *Diagnostics* | |
| $\rho(u)$ | Leakage pressure (outside/total energy ratio). |

Table 3 – continued from previous page

| Symbol | Description |
|---|---|
| $\rho_k$ | Discrete leakage pressure at step $k$. |
| $\omega(u)$ | Scale-wise directional oscillation. |
| $\omega_k$ | Step-to-step oscillation on $\Delta V_k^{\text{eff}}$. |
| $\delta$ | Small scale offset (for oscillation). |
| $\cos(a, b)$ | Cosine similarity. |
| $\langle a, b \rangle$ | Inner product. |
| $\|\cdot\|_2$ | $\ell_2$ norm. |
| $\varepsilon$ | Numerical stabilizer (e.g., $10^{-8}$). |
| *Rollout functional and proxy* | |
| $\phi(x, u)$ | Local nonnegative risk density. |
| $\phi_k$ | Step risk density (proxy instantiation). |
| $\lambda_\rho, \lambda_\omega, \lambda_h$ | Weights in $\phi_k$. |
| $\mathcal{G}[x(\cdot), u(\cdot)]$ | Rollout functional, $\int_0^1 \phi(\cdot)\, ds$. |
| $\widehat{\mathcal{G}}$ | Discrete rollout proxy, $\sum_k \Delta s_k\, \phi_k$. |
| $m_{\text{bad}}$ | Outside-window progress mass. |
| $\mathbf{1}\{\cdot\}$ | Indicator function. |
| *Scale regimes and schedule families* | |
| $\mathcal{U}_{\text{eff}}$ | Effective scale window (tail window). |
| $\mathcal{U}_{\text{good}}$ | Low-risk scale interval. |
| $\mathcal{U}_{\text{bad}}$ | High-risk scale interval. |
| $\mathcal{U}^\star$ | Good sub-window used in theory. |
| $\mathcal{F}_{\text{couple}}(K)$ | Coupled budget-to-range schedule family. |
| $\mathcal{F}_{\text{decouple}}(K)$ | Decoupled (density-refinement) schedule family. |
| $T_{\text{bad}}$ | Preimage set of bad scales (in proofs). |
| $t_{\text{bad}}$ | Bad cutpoint (in proofs). |
| $t_0(K)$ | Coupled suffix start parameter (budget-dependent). |
| $\delta(K)$ | Lower bound on forced bad mass (proof bound). |
| $c_{\text{good}}, c_{\text{bad}}$ | Risk bounds on good/bad regimes. |
| *ODE / numerical analysis (appendix)* | |
| $f(s, x)$ | ODE vector field (proof notation). |
| $g(s)$ | Scale-to-progress factor (proof notation). |
| $s_k$ | Grid point on $s$ for Euler discretization. |
| $h_s$ | Euler step size on $s$ (typically $1/K$). |
| $h_u$ | Max scale step size (density measure). |
| $x^\star$ | Reference/ideal trajectory (proof notation). |
| $u^\star$ | Window-fixed control (proof notation). |
| $x_k^{(K)}$ | Euler discrete iterate (proof notation). |
| $L$ | Lipschitz constant (for $f$). |
| $B$ | Bound on $\|f\|$ (proof). |
| $L_\phi$ | Lipschitz constant (for $\phi$). |
| $C_{\text{E}}$ | Euler global error constant. |
| $W_u$ | scale-span width. |
| *Evaluation metrics (tables/figures)* | |
| Dist | Structure distance metric. |
| PSNR | Peak signal-to-noise ratio. |
| MSE | Mean squared error. |
| SSIM | Structural similarity index. |
| LPIPS | Perceptual distance (LPIPS). |

| Table 3 – continued from previous page | |
| --- | --- |
| **Symbol** | **Description** |
| $\mathrm{PSNR}_{\mathrm{bg}}$ | Background PSNR (background fidelity). |
| CLIP (Whole/Edited) | CLIP semantic scores (whole image / edited region). |

# C. Instantiation of $\phi$ and $\widehat{\mathcal{G}}$

This section specifies the concrete rollout proxy used in Figure 5 and Figure 6. The design goal is purely operational: $\widehat{\mathcal{G}}$ should be computable from signals already produced during editing (no extra model calls), and should reflect the two weak premises used in the main text: $\phi$ increases under spatial underdetermination (leakage), and the discretization component of the rollout decreases when density is refined inside a fixed effective window.

## C.1. Per-step signals available during editing

Fix a run with $K$ model evaluations and a monotone scale sequence $\{u_k\}_{k=0}^{K}$. Each step constructs the co-located pair $z_k^{\mathrm{src}} = (1 - u_k)x_{\mathrm{src}} + u_k \epsilon_k$, $z_k^{\mathrm{tar}} = x_k + (z_k^{\mathrm{src}} - x_{\mathrm{src}})$, queries the model once (source and target in a batch), and forms the differential field

$$\Delta V_k = v_\theta(z_k^{\mathrm{tar}}, \tau(u_k), c_{\mathrm{tar}}) - v_\theta(z_k^{\mathrm{src}}, \tau(u_k), c_{\mathrm{src}}).$$

Let $M_k \in [0, 1]^{H \times W \times C}$ denote the feasible-region mask extracted from the model internal representation at the same query (main text, Sec. 3.1). Define the masked (effective) differential field

$$\Delta V_k^{\mathrm{eff}} := M_k \odot \Delta V_k.$$

We use two diagnostics.

**Leakage pressure.** With a small $\varepsilon > 0$,

$$\rho_k := \frac{\|(1 - M_k) \odot \Delta V_k\|_2}{\|\Delta V_k\|_2 + \varepsilon}. \tag{4}$$

This is exactly Eq. (1) instantiated at $u_k$.

**Directional oscillation.** Let $\cos(a, b) = \langle a, b \rangle / ((\|a\|_2 + \varepsilon)(\|b\|_2 + \varepsilon))$. We measure step-to-step instability on the effective field:

$$\omega_k := \begin{cases} 0, & k = 0, \\ 1 - \cos(\Delta V_k^{\mathrm{eff}}, \Delta V_{k-1}^{\mathrm{eff}}), & k \geq 1. \end{cases} \tag{5}$$

## C.2. Progress weights and outside-window mass

The rollout functional in the main text integrates over the progress axis $s \in [0, 1]$. In the self-consistent discretization (Theorem 4.4), progress increments are induced by the scale increments. We therefore define per-step progress weights from the scale steps:

$$\Delta u_k := u_{k+1} - u_k, \qquad \Delta s_k := \frac{|\Delta u_k|}{\sum_{j=0}^{K-1} |\Delta u_j|}, \qquad \sum_{k=0}^{K-1} \Delta s_k = 1. \tag{6}$$

Given an effective window $\mathcal{U}_{\mathrm{eff}}$ (defined in the main text by the fixed tail window on the scheduler path), the outside-window progress mass is

$$m_{\mathrm{bad}} := \sum_{k=0}^{K-1} \Delta s_k \, \mathbf{1}\{u_k \notin \mathcal{U}_{\mathrm{eff}}\}. \tag{7}$$

## C.3. Operational effective window $\mathcal{U}_{\mathrm{eff}}$ from a tail window

NaviEdit does not recompute $\mathcal{U}_{\mathrm{eff}}$ per instance. Instead, it operationalizes $\mathcal{U}_{\mathrm{eff}}$ by selecting a fixed tail window on the scheduler path and restricting visited scales to that suffix. Concretely, let $\{(t^{(n)}, u^{(n)})\}_{n=0}^{N-1}$ be the monotone path from

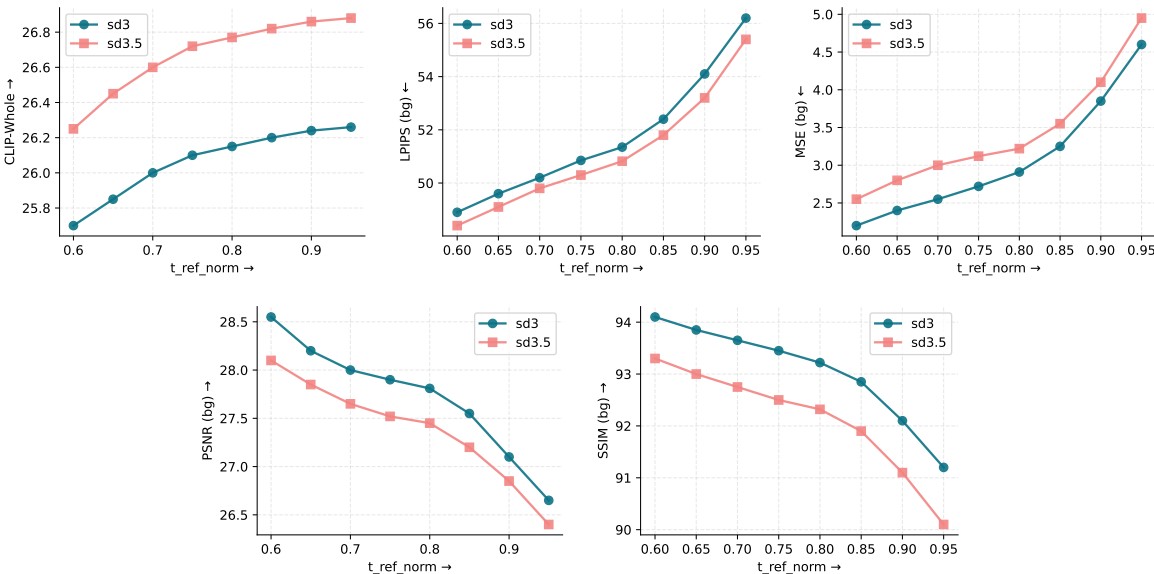

*Figure 12.* **Sensitivity to the operational effective window.** Sweeping $t_{\text{ref}}$ varies the tail-window width; the qualitative density-versus-range trend remains stable.

noisy→clean used by the scheduler. We choose a reference depth $t_{\text{ref}}$ and set the tail start index as $n_0 := \max(0, N - t_{\text{ref}})$, so the visited interval is

$$\mathcal{U}_{\text{eff}} := \{u^{(n)} : n \in \{n_0, \ldots, N - 1\}\}.$$

Unless stated otherwise, the deployed PIE-Bench Navi-FlowEdit + gate system uses a 50-step scheduler grid ($N = 50$), a 50-step editing budget ($K = 50$), and $t_{\text{ref}} = 42$. Other budget settings such as the $K = 20/28$ step-scaling study are controlled ablations and should not be read as the deployed Table 1 setting.

### C.4. Sensitivity to $t_{\text{ref}}$

We sweep $t_{\text{ref}}$ to vary the width of $\mathcal{U}_{\text{eff}}$ while keeping the step budget fixed. This directly tests how performance depends on the operational window choice and supports the use of a fixed tail window instead of per-instance oracle selection.

### C.5. Local risk density $\phi$ and rollout proxy $\widehat{\mathcal{G}}$

We instantiate a nonnegative per-step risk density as a fixed-weight combination of leakage, oscillation, and a discretization term:

$$\phi_k := \lambda_\rho \, \rho_k + \lambda_\omega \, \omega_k + \lambda_h \, |\Delta u_k|. \tag{8}$$

We use fixed weights across all experiments: $(\lambda_\rho, \lambda_\omega, \lambda_h) = (1.0, 1.0, 0.05)$. We set $\varepsilon = 10^{-8}$ in all cosine/ratio computations.

The discrete rollout proxy is the progress-weighted accumulation:

$$\widehat{\mathcal{G}} := \sum_{k=0}^{K-1} \Delta s_k \, \phi_k. \tag{9}$$

When we report $\widehat{\mathcal{G}}$ for a schedule family, we average Eq. (9) over the evaluation set and random seeds (fresh $\epsilon_k$ per step).

### C.6. Why this instantiation matches the two weak premises

**Sensitivity to underdetermination.** By construction, $\rho_k$ increases when the prompt-induced motion leaks outside the feasible region. Since $\phi_k$ is monotone in $\rho_k$, steps at underdetermined scales yield larger local risk contributions.

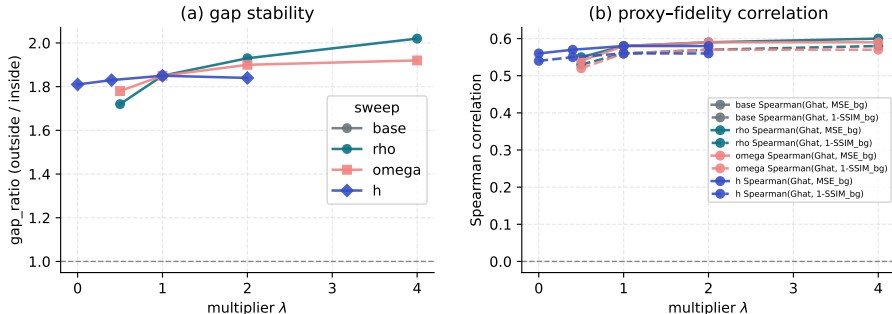

*Figure 13.* **Sensitivity of the instantiated proxy.** Sweeping $\lambda_\rho, \lambda_\omega, \lambda_h$ shows that the qualitative trends used in the paper are stable within a reasonable range of weights.

**Density refinement decreases the discretization component inside a fixed window.** Consider controls that stay inside a fixed window $\mathcal{U}^\star$ and differ only by density (i.e., by how the same total scale span is partitioned into steps). The discretization part of Eq. (9) is

$$\widehat{\mathcal{G}}_{\mathrm{disc}} = \sum_{k=0}^{K-1} \Delta s_k \, \lambda_h |\Delta u_k| = \lambda_h \frac{\sum_{k=0}^{K-1} |\Delta u_k|^2}{\sum_{j=0}^{K-1} |\Delta u_j|}.$$

Let $W_u := \sum_{j=0}^{K-1} |\Delta u_j|$ be the total traversed scale span (fixed by the window endpoints). Then

$$\widehat{\mathcal{G}}_{\mathrm{disc}} = \lambda_h \frac{\sum_k |\Delta u_k|^2}{W_u} \le \lambda_h \frac{(\max_k |\Delta u_k|) \sum_k |\Delta u_k|}{W_u} = \lambda_h \max_k |\Delta u_k|. \tag{10}$$

Thus, at fixed $W_u$, refining density (decreasing $\max_k |\Delta u_k|$) cannot increase the discretization component and typically decreases it. This is the only property of discretization used by the main text claims: density refinement inside an effective window reduces the rollout proxy through the step-size term, while range expansion can introduce additional leakage/instability through $\rho_k$ and $\omega_k$ as the trajectory enters risky regimes.

### C.7. Sensitivity to $(\lambda_\rho, \lambda_\omega, \lambda_h)$

We evaluate robustness to the proxy instantiation by sweeping one weight at a time while keeping the other two fixed. We report the induced changes in $\widehat{\mathcal{G}}$ and the resulting editability–fidelity trade-off under the same fixed step budget.

### C.8. Timestep mapping $\tau(u)$

Let $\{t^{(n)}\}_{n=0}^{N-1}$ be the scheduler timesteps used at inference. We define the normalized scale as $u^{(n)} = t^{(n)}/T$, where $T$ is the scheduler's training-time timestep scale (e.g., `num_train_timesteps`). Accordingly, we use the linear map $\tau(u) = u\,T$, so that $\tau(u^{(n)}) = t^{(n)}$. In practice we maintain a monotone path $(t_{\mathrm{path}}[n], u_{\mathrm{path}}[n])$ and interpolate with a shared $p \in [0, 1]$ to obtain $(t, u)$, ensuring that mixing, querying, and actuation share the same scale.

## D. Proof of Theorem 4.2

*Proof.* We make explicit the two ingredients used by the theorem: (i) a *scale-regime gap* in the local risk density $\phi$, and (ii) a *coupling-induced* lower bound on how much progress must be spent in the bad regime.

**Step 0 (Notation).** For any admissible control $u(\cdot)$, define the fraction of progress spent in the bad regime

$$m_{\mathrm{bad}}\big(u(\cdot)\big) := \int_0^1 \mathbf{1}\{u(s) \in \mathcal{U}_{\mathrm{bad}}\}\, ds \in [0, 1]. \tag{11}$$

Write $\mathcal{G}[u(\cdot)]$ as shorthand for $\mathcal{G}[x(\cdot), u(\cdot)]$ where $x(\cdot)$ is the induced trajectory under Definition 4.1.

**Step 1 (Risk gap assumption, made explicit).** By the premise of Theorem 4.2, there exist constants

$$0 \le c_{\mathrm{good}} < c_{\mathrm{bad}} \tag{12}$$

such that, for all states $x$ reachable under the considered instance,

$$\phi(x, u) \leq c_{\text{good}} \quad \text{for } u \in \mathcal{U}_{\text{good}}, \qquad \phi(x, u) \geq c_{\text{bad}} \quad \text{for } u \in \mathcal{U}_{\text{bad}}. \tag{13}$$

We additionally use the explicit separation condition

$$c_{\text{bad}} \delta(K) > c_{\text{good}}, \tag{14}$$

where $\delta(K)$ is the lower bound on the bad-regime progress mass enforced by the coupled family for the considered budgets.

**Step 2 (A valid lower bound for any control that visits the bad regime).** Using the nonnegativity of $\phi$ and its lower bound on $\mathcal{U}_{\text{bad}}$,

$$
\begin{aligned}
\mathcal{G}[u(\cdot)] &= \int_0^1 \phi\big(x(s), u(s)\big)\, ds \\
&\geq \int_{\{s:\, u(s) \in \mathcal{U}_{\text{bad}}\}} \phi\big(x(s), u(s)\big)\, ds \\
&\geq c_{\text{bad}} \int_0^1 \mathbf{1}\{u(s) \in \mathcal{U}_{\text{bad}}\}\, ds \\
&= c_{\text{bad}}\, m_{\text{bad}}(u(\cdot)).
\end{aligned}
\tag{15}
$$

**Step 3 (Coupling implies a bad-mass lower bound).** By the theorem assumption, for the considered budgets,

$$m_{\text{bad}}(u(\cdot)) \geq \delta(K) > 0, \qquad \forall\, u(\cdot) \in \mathcal{F}_{\text{couple}}(K). \tag{16}$$

**Step 4 (Lower bound for the coupled family).** Combining (15) with (16) yields, for the considered budgets,

$$\inf_{u(\cdot) \in \mathcal{F}_{\text{couple}}(K)} \mathcal{G}[u(\cdot)] \;\geq\; c_{\text{bad}}\, \delta(K). \tag{17}$$

**Step 5 (A decoupled control with smaller objective).** By the decoupled-feasibility assumption, choose $u^\star(\cdot) \in \mathcal{F}_{\text{decouple}}(K)$ such that $u^\star(s) \in \mathcal{U}_{\text{good}}$ for all $s \in [0, 1]$. By (13),

$$\mathcal{G}[u^\star(\cdot)] = \int_0^1 \phi\big(x^\star(s), u^\star(s)\big)\, ds \;\leq\; \int_0^1 c_{\text{good}}\, ds = c_{\text{good}}, \tag{18}$$

hence

$$\inf_{u(\cdot) \in \mathcal{F}_{\text{decouple}}(K)} \mathcal{G}[u(\cdot)] \;\leq\; c_{\text{good}}. \tag{19}$$

**Step 6 (Strict gap).** Combining (17) and (19) gives, for the considered budgets,

$$\inf_{u(\cdot) \in \mathcal{F}_{\text{couple}}(K)} \mathcal{G}[u(\cdot)] - \inf_{u(\cdot) \in \mathcal{F}_{\text{decouple}}(K)} \mathcal{G}[u(\cdot)] \geq c_{\text{bad}} \delta(K) - c_{\text{good}} \;>\; 0, \tag{20}$$

where the last inequality follows from the separation condition in (14). This establishes the incompleteness of the coupled family: the coupled budget-to-range constraint enforces a sufficiently costly progress mass in $\mathcal{U}_{\text{bad}}$, while the decoupled family can allocate progress inside $\mathcal{U}_{\text{good}}$, yielding a strictly smaller attainable rollout functional value. $\qquad\square$

### D.1. Coupled vs. decoupled schedules under increasing step budget

We evaluate how increasing the practical step budget changes the fidelity–editability tradeoff under two schedule families that differ only in scale allocation. The coupled family follows the common heuristic where increasing the budget effectively expands the accessed scale range toward noisier regimes, whereas the decoupled family keeps a fixed effective window

*Table 4.* Effect of increasing step budget under coupled vs. decoupled scale allocation on SD3 with the gate-enabled Navi-FlowEdit configuration held fixed. Coupled schedules expand into noisier scales as the budget increases, while decoupled schedules densify within a fixed effective window. The FlowEdit row is a matched-step controlled reference; its official-step result is reported in Table 1.

| Method | Schedule | Dist↓ | PSNR↑ | MSE↓ | SSIM↑ | LPIPS↓ | CLIP-Whole↑ | CLIP-Edit↑ | Steps | Runtime(s)↓ | VRAM(MiB)↓ |
|---|---|---|---|---|---|---|---|---|---|---|---|
| Navi-FlowEdit + gate | Coupled | 14.28 | 26.12 | 4.75 | 90.10 | 71.29 | 24.12 | 21.35 | 20 | 14.95 | 22120 |
| | Decoupled | 10.84 | 27.92 | 2.82 | 93.61 | 48.62 | 24.51 | 21.62 | 20 | 14.95 | 22120 |
| | Coupled | 27.22 | 22.18 | 9.61 | 88.22 | 87.98 | 26.01 | 22.55 | 28 | 17.52 | 22120 |
| | Decoupled | 11.10 | 27.81 | 2.91 | 93.22 | 51.35 | 26.15 | 22.67 | 28 | 17.52 | 22120 |
| FlowEdit (matched-step reference) | Coupled | 17.06 | 21.52 | 8.33 | 86.09 | 129.54 | 25.80 | 22.42 | 20 | 14.55 | 17140 |

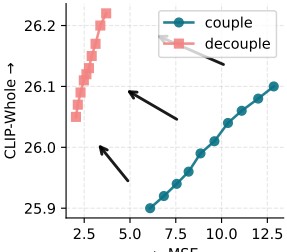 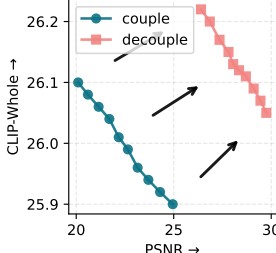

*Figure 14.* **Decoupling gains a better balance under a matched step budget.** We plot CLIP-Whole (edit compliance) against background fidelity for coupled and decoupled schedule families under the same model and evaluation budget. Decoupled schedules consistently attain higher PSNR at comparable CLIP-Whole, indicating a family-level frontier shift rather than a single tuned operating point.

and uses additional steps to densify sampling within that window. All other components are held fixed, including model, guidance, gating, and self-consistent mixing/querying/update axes.

For Navi-FlowEdit, the step budget coincides with the number of model evaluations in our controller. For FlowEdit, we report matched *editing steps* rather than a literal model-evaluation count, because the official implementation uses a different scheduler/accounting convention. Increasing the budget from 20 to 28 improves edit alignment for both schedule families (higher CLIP-Edit), but only the coupled schedule suffers a pronounced fidelity collapse (lower PSNR/SSIM and higher Dist/MSE/LPIPS). In contrast, the decoupled schedule preserves fidelity while retaining comparable edit strength. This isolates the scale-allocation effect in Theorems 4.2 and 4.3: allocating additional budget to range expansion forces nontrivial mass into high-risk scales, whereas allocating it to density refinement within the effective window improves numerical stability without sacrificing editability.

# E. Proof of Theorem 4.3

*Proof.* We prove the claim for the finite-budget quantity controlled by inference. All statements are conditioned on the fixed anchor realization specified in Theorem 4.3.

For a $K$-step self-consistent rollout $\{(x_k, u_k)\}_{k=0}^{K}$, let $\Delta s_k \geq 0$ be normalized progress weights with $\sum_{k=0}^{K-1} \Delta s_k = 1$. Define the implemented rollout cost and implemented bad-regime mass as

$$\mathcal{G}_K[x_{0:K}, u_{0:K}] := \sum_{k=0}^{K-1} \Delta s_k \, \phi(x_k, u_k), \qquad m_{\text{bad}}^K := \sum_{k=0}^{K-1} \Delta s_k \, \mathbf{1}\{u_k \in \mathcal{U}_{\text{bad}}\}. \tag{21}$$

**Range-expanding rollouts.** For any range-expanding rollout satisfying $m_{\text{bad}}^K \geq \delta_K$, the assumptions $\phi \geq 0$ and $\phi(x, u) \geq c_{\text{bad}}$ on $\mathcal{U}_{\text{bad}}$ give

$$\mathcal{G}_K[x_{0:K}, u_{0:K}] = \sum_{k=0}^{K-1} \Delta s_k \, \phi(x_k, u_k)$$

$$\geq \sum_{k=0}^{K-1} \Delta s_k \, c_{\text{bad}} \mathbf{1}\{u_k \in \mathcal{U}_{\text{bad}}\}$$

$$= c_{\text{bad}} m_{\text{bad}}^K \geq c_{\text{bad}} \delta_K. \tag{22}$$

Thus every range-expanding rollout in the considered family has implemented cost at least $c_{\mathrm{bad}}\delta_K$.

**Density refinement inside the good window.** Consider the self-consistent density-refined rollout inside $\mathcal{U}^\star \subseteq \mathcal{U}_{\mathrm{good}}$ from the theorem. Let $u^\star(s)$ be its continuous good-window control and $x^\star(s)$ the corresponding ideal trajectory. Let $x_k^{(K)}$ be the $K$-step self-consistent Euler rollout on the grid $s_k = k/K$ with $h_s = 1/K$:

$$x_{k+1}^{(K)} = x_k^{(K)} + \left(u^\star(s_{k+1}) - u^\star(s_k)\right)\Delta V_{\mathrm{eff}}\left(x_k^{(K)}; u^\star(s_k), \epsilon_k\right). \tag{23}$$

By the theorem assumption,

$$\max_{0 \le k \le K}\left\|x_k^{(K)} - x^\star(s_k)\right\| \le \frac{C_{\mathrm{E}}}{K}. \tag{24}$$

Since $u^\star(s_k) \in \mathcal{U}^\star$ and $\phi(x, u) \le c_{\mathrm{good}}$ on $\mathcal{U}^\star$, while $\phi$ is $L_\phi$-Lipschitz in $x$ there, the implemented cost of this density-refined rollout satisfies

$$
\begin{aligned}
\mathcal{G}_K^{\mathrm{dens}} &:= \sum_{k=0}^{K-1} h_s\, \phi\left(x_k^{(K)}, u^\star(s_k)\right) \\
&\le \sum_{k=0}^{K-1} h_s\, \phi(x^\star(s_k), u^\star(s_k)) + \sum_{k=0}^{K-1} h_s L_\phi \left\|x_k^{(K)} - x^\star(s_k)\right\| \\
&\le c_{\mathrm{good}} + L_\phi \max_k \left\|x_k^{(K)} - x^\star(s_k)\right\| \\
&\le c_{\mathrm{good}} + \frac{L_\phi C_{\mathrm{E}}}{K}.
\end{aligned}
\tag{25}
$$

**Strict comparison.** By the margin assumption in Theorem 4.3,

$$c_{\mathrm{bad}}\delta_K - c_{\mathrm{good}} \ge \gamma > 0. \tag{26}$$

If $K > L_\phi C_{\mathrm{E}}/\gamma$, then

$$c_{\mathrm{good}} + \frac{L_\phi C_{\mathrm{E}}}{K} < c_{\mathrm{good}} + \gamma \le c_{\mathrm{bad}}\delta_K. \tag{27}$$

Combining (22) and (25) yields

$$\mathcal{G}_K^{\mathrm{dens}} < \inf_{\substack{K\text{-step range-expanding rollouts} \\ m_{\mathrm{bad}}^K \ge \delta_K}} \mathcal{G}_K. \tag{28}$$

Hence, under the stated assumptions and fixed budget $K$, self-consistent density refinement inside $\mathcal{U}^\star$ attains lower implemented rollout cost than any range-expanding rollout that allocates at least $\delta_K$ progress mass to $\mathcal{U}_{\mathrm{bad}}$. $\qquad\square$

## F. Proof of Theorem 4.4

*Proof.* We prove the two directions in Theorem 4.4. Throughout, fix a run with a monotone scale sequence $\{u_k\}_{k=0}^K$ and fresh anchors $\epsilon_k \sim \mathcal{N}(0, I)$. Write $\Delta u_k = u_{k+1} - u_k$. For any scale $u$, define the co-located anchors (as in Sec. 3.1)

$$z^{\mathrm{src}}(x_{\mathrm{src}}; u, \epsilon) = (1 - u)x_{\mathrm{src}} + u\epsilon, \qquad z^{\mathrm{tar}}(x; x_{\mathrm{src}}; u, \epsilon) = x + \left(z^{\mathrm{src}}(x_{\mathrm{src}}; u, \epsilon) - x_{\mathrm{src}}\right),$$

and the corresponding effective differential field

$$\Delta V_{\mathrm{eff}}(x; u, \epsilon) := \Pi_{M(u)}\left(v_\theta\left(z^{\mathrm{tar}}(x; x_{\mathrm{src}}; u, \epsilon), \tau(u), c_{\mathrm{tar}}\right) - v_\theta(z^{\mathrm{src}}(x_{\mathrm{src}}; u, \epsilon), \tau(u), c_{\mathrm{src}})\right), \tag{29}$$

where $\Pi_{M(u)}$ denotes the active-set projection induced by the feasible-region mask at that query (mask extraction details are irrelevant to this proof; we only use that $\Pi_{M(u)}$ is well-defined for the same query state).

**Step 0 (What it means to be a consistent discretization).** Definition 4.1 states that editing is a controlled ODE on the progress axis $s$:

$$\frac{dx}{ds} = \frac{du}{ds}\,\Delta V_{\mathrm{eff}}\left(x(s); u(s), \epsilon(s)\right). \tag{30}$$

A first-order (explicit Euler) discretization of (30) on a grid $0 = s_0 < \cdots < s_K = 1$ reads

$$x_{k+1} = x_k + \Delta s_k \frac{du}{ds}(s_k) \, \Delta V_{\text{eff}}(x_k; u(s_k), \epsilon(s_k)), \qquad \Delta s_k := s_{k+1} - s_k. \tag{31}$$

If we choose the progress parameterization so that $s$ is proportional to the traversed scale coordinate, namely

$$\Delta s_k = \frac{\Delta u_k}{u(1) - u(0)} \quad \text{for monotone } u(\cdot), \tag{32}$$

then $\Delta s_k \frac{du}{ds}(s_k) = \Delta u_k$ and (31) becomes

$$x_{k+1} = x_k + \Delta u_k \, \Delta V_{\text{eff}}(x_k; u_k, \epsilon_k), \tag{33}$$

which is exactly the update stated in the theorem. Thus, the "consistent discretization" claim reduces to checking that the discrete step uses the *same* scale coordinate $u_k$ to (i) define the anchors, (ii) query the model through $\tau(u_k)$, and (iii) scale the update by $\Delta u_k$.

**Step 1 (Consistency when the step is self-consistent).** Assume the step is self-consistent in the sense of the theorem: it uses $u_k$ for mixing to form $z_k^{\text{src}} = z^{\text{src}}(x_{\text{src}}; u_k, \epsilon_k)$, uses the same $u_k$ for the query time $\tau(u_k)$ and mask $\Pi_{M(u_k)}$, and uses $\Delta u_k$ as the multiplier of the differential field computed at $(x_k, u_k, \epsilon_k)$. By definition (29), the computed increment is exactly $\Delta V_{\text{eff}}(x_k; u_k, \epsilon_k)$. Therefore the implemented update is (33), which is the explicit Euler step of the controlled ODE (30) under the progress parameterization (32). This proves the second part of Theorem 4.4.

**Step 2 (Inconsistency under axis mismatch: no ODE can generate the same step).** We now consider a mismatched step where the three occurrences of the scale coordinate disagree. To keep the statement minimal and aligned with the ablations in Sec. 4.5, it suffices to analyze a single mismatch mode; the same argument applies to the others.

*Mismatch-query.* Suppose mixing and update use $u_k$ and $\Delta u_k$, but the model is queried at $\tau(\tilde{u}_k)$ for some $\tilde{u}_k \neq u_k$. Then the implemented update is

$$x_{k+1} = x_k + \Delta u_k \, \widetilde{\Delta V}_{\text{eff}}(x_k; u_k, \tilde{u}_k, \epsilon_k), \tag{34}$$

where $\widetilde{\Delta V}_{\text{eff}}$ denotes the differential field formed by anchors constructed at $u_k$ but queried at $\tilde{u}_k$:

$$\widetilde{\Delta V}_{\text{eff}}(x; u, \tilde{u}, \epsilon) := \Pi_{M(\tilde{u})}\big(v_\theta\big(z^{\text{tar}}(x; x_{\text{src}}; u, \epsilon), \tau(\tilde{u}), c_{\text{tar}}\big) - v_\theta(z^{\text{src}}(x_{\text{src}}; u, \epsilon), \tau(\tilde{u}), c_{\text{src}})\big).$$

Compare this with the self-consistent Euler step (33):

$$x_{k+1}^{\text{sc}} = x_k + \Delta u_k \, \Delta V_{\text{eff}}(x_k; u_k, \epsilon_k).$$

The mismatch introduces an additive deviation

$$b_k := \Delta u_k \left( \widetilde{\Delta V}_{\text{eff}}(x_k; u_k, \tilde{u}_k, \epsilon_k) - \Delta V_{\text{eff}}(x_k; u_k, \epsilon_k) \right), \tag{35}$$

so that $x_{k+1} = x_{k+1}^{\text{sc}} + b_k$.

We claim that, unless $\tilde{u}_k = u_k$, the update (34) cannot be written as a consistent discretization of Definition 4.1 for the *same* control input $u(\cdot)$. Indeed, in Definition 4.1, the right-hand side at progress time $s_k$ is determined by the triple

$$(\text{anchors at } u(s_k), \quad \text{query at } \tau(u(s_k)), \quad \text{actuation by } du/ds(s_k)),$$

all sharing the *same* scale value $u(s_k)$. A first-order consistent discretization at grid point $s_k$ must therefore use (33) with the same $u_k$ for anchors and query time. But (34) uses anchors at $u_k$ and query time at $\tilde{u}_k \neq u_k$, which is not of the Euler form of (30) under any choice of $du/ds$ at $s_k$ because $du/ds$ only rescales the vector field evaluated at $u_k$; it cannot change the vector field evaluation from $\tau(u_k)$ to $\tau(\tilde{u}_k)$.

Equivalently, suppose for contradiction that (34) were a consistent discretization of some instance of (30) with control $u(\cdot)$ satisfying $u(s_k) = u_k$. Then the one-step increment divided by $\Delta u_k$ would have to equal $\Delta V_{\text{eff}}(x_k; u_k, \epsilon_k)$, which is defined by querying at $\tau(u_k)$. But the actual increment divided by $\Delta u_k$ equals $\widetilde{\Delta V}_{\text{eff}}(x_k; u_k, \tilde{u}_k, \epsilon_k)$ which queries at

$\tau(\tilde{u}_k)$. These are different functions of $(x_k, \epsilon_k)$ whenever $\tilde{u}_k \neq u_k$ unless the model outputs are scale-invariant in $t$ (which they are not in general, and would contradict the empirical regime structure in Sec. 3.2). Therefore no controlled ODE of the form in Definition 4.1 with the same control $u(\cdot)$ can yield (34) as its first-order consistent step.

**Step 3 (The systematic bias term and why it accumulates).** Equation (35) gives an explicit bias term $b_k$ measuring the discrepancy between what is *measured* (the differential field under one scale) and what is *applied* (an update step sized by another scale increment). This term is systematic because it is induced deterministically by the mismatch pattern, not by stochasticity in $\epsilon_k$ (fresh noise is still used).

When such a bias persists over steps, it accumulates as

$$x_K = x_0 + \sum_{k=0}^{K-1} \Delta u_k \, \Delta V_{\text{eff}}(x_k; u_k, \epsilon_k) + \sum_{k=0}^{K-1} b_k, \tag{36}$$

so the mismatch behaves like an uncontrolled forcing term on top of the intended controlled dynamics. In practice this manifests as drift, artifacts, and eventual collapse into a generative mode, which is exactly what the axis-mismatch ablations in Figure 7 are designed to test.

**Step 4 (Other mismatch modes).** The same structure applies to mismatch-mix (anchors built at $\tilde{u}_k$ but query/update at $u_k$) and mismatch-step (update scaled by $\Delta \tilde{u}_k$ while measuring at $u_k$): each can be written as the self-consistent Euler increment plus an additive deviation of the form (35), induced by evaluating different components of the step on different scale coordinates. Thus, in all mismatch cases, the step is not a consistent discretization of Definition 4.1 and incurs a systematic bias term.

This proves Theorem 4.4. □

# G. Optional Internal Feasible-Region Gate in Navi-FlowEdit

## G.1. Internal feasible-region mask extraction (no external masks)

The internal gate is an optional component used in Navi-FlowEdit to suppress off-region drift. It is not required by the Navi controller itself, and variants such as Navi-FlowAlign are run with $M \equiv 1$. For Navi-FlowEdit, we compute the feasible-region gate $M_k$ from internal signals produced by the same model call used to obtain $\Delta V_k$, and never require user-provided masks. At step $k$, we form the co-located pair $(z_k^{\text{src}}, z_k^{\text{tar}})$ and query the model once with a concatenated batch (source and target, optionally with CFG). Let $h^{(\ell)}(\cdot) \in \mathbb{R}^{N \times C}$ denote the token-wise hidden states emitted by transformer block $\ell$ for that same forward pass, with $N = H \cdot W$.

We use a training-free representation-difference heatmap:

$$A_k := \frac{1}{|\mathcal{L}|} \sum_{\ell \in \mathcal{L}} \text{reshape}_{H \times W} \left( \frac{1}{C} \left\| h^{(\ell)}(z_k^{\text{tar}}) - h^{(\ell)}(z_k^{\text{src}}) \right\|_2^2 \right) \in \mathbb{R}^{H \times W}, \tag{37}$$

where $\mathcal{L}$ is a fixed set of blocks chosen once for the model. In our implementation, $\mathcal{L}$ is the middle third of transformer blocks (from $\lfloor L/3 \rfloor$ to $\lfloor 2L/3 \rfloor - 1$), which provides stable localization while avoiding early high-noise features and late texture-dominated features.

We then convert $A_k$ into a soft gate $M_k \in [0,1]^{H \times W}$ using a small sequence of deterministic operations:

$$\tilde{A}_k := \text{NormMinMax}(A_k), \tag{38}$$

$$\bar{A}_k := \text{AvgPool}(\tilde{A}_k; k_{\text{blur}}), \tag{39}$$

$$\hat{M}_k := \mathbf{1}\{\bar{A}_k \geq \text{Quantile}(\bar{A}_k, q)\}, \tag{40}$$

followed by area control and temporal smoothing:

$$\hat{M}_k \leftarrow \text{AreaBound}(\hat{M}_k; a_{\min}, a_{\max}), \tag{41}$$

$$M_k \leftarrow \beta M_{k-1} + (1-\beta)\hat{M}_k, \qquad M_k \in [0,1]^{H \times W}. \tag{42}$$

*Table 5.* Compute overhead of internal masking on SD3 (PIE-Bench, bs=1, 1024×1024, 20 edit steps, same CFG and precision). Extra model calls are always zero because masking reuses the same model query. VRAM reports peak allocated memory.

| Mask setting | Extra model calls | Time/img (s)↓ | VRAM (MiB)↓ |
|---|---|---|---|
| $M \equiv 1$ (mask disabled) | 0 | 14.55 | 17140 |
| Internal mask (ours) | 0 | 14.95 | 22120 |

Default hyperparameters (used throughout unless stated otherwise) are: $q = 0.90$ (keep the top $10\%$ salient region), $a_{\min} = 0.02$ and $a_{\max} = 0.65$ (area fraction bounds), $k_{\mathrm{blur}} = 5$ (average pooling kernel), and $\beta = 0.85$ (EMA). We additionally apply a scale-aware dilation to reduce brittleness at noisier scales:

$$M_k \leftarrow \mathrm{MaxPool}(M_k; k_{\mathrm{dil}}(u_k)), \qquad k_{\mathrm{dil}}(u_k) = 1 + \mathrm{round}\big((k_{\max} - 1)\, u_k\big), \tag{43}$$

with $k_{\max} = 7$ and odd kernel sizes.

If model hooks are unavailable, we fall back to a $\Delta V$-saliency mask:

$$A_k^{\Delta V} := \frac{1}{C}\|\Delta V_k\|_2^2 \in \mathbb{R}^{H \times W}, \tag{44}$$

followed by the same normalization, blur, quantile thresholding, area control, and EMA.

Finally, the gate is used only to form the effective differential field

$$\Delta V_k^{\mathrm{eff}} := (M_k^\gamma) \odot \Delta V_k, \tag{45}$$

with $\gamma = 1.0$ by default. We do not apply any hard replacement outside the mask unless explicitly enabled.

### G.2. Active-set growth

The gate above is designed to be conservative; however, overly small gates can stall semantic motion. We therefore allow a lightweight active-set growth rule driven by a pressure ratio computed from already-available tensors:

$$r_k := \frac{\|(1 - M_k) \odot \Delta V_k\|_2}{\|\Delta V_k\|_2 + \varepsilon}. \tag{46}$$

If $r_k > \tau_r$ for $P$ consecutive steps, we expand $M_k$ by adding a small set of pixels with highest outside-mask energy:

$$P_k := \mathrm{NormMinMax}\left(\frac{1}{C}\|(1 - M_k) \odot \Delta V_k\|_2^2\right), \tag{47}$$

$$\Delta M_k := \mathbf{1}\{P_k \geq \mathrm{Quantile}(P_k, q_g)\}, \tag{48}$$

$$M_k \leftarrow \max(M_k, \mathrm{MaxPool}(\Delta M_k; k_g)), \tag{49}$$

followed by the same area control. Defaults are $\tau_r = 0.35$, $P = 2$, $q_g = 0.97$, and $k_g = 3$.

### G.3. Compute overhead of internal masking

Mask extraction does not introduce additional model evaluations: $M_k$ is computed from tensors produced by the same forward pass used to compute $\Delta V_k$. It does introduce a small constant-factor overhead from per-step tensor operations (pooling, quantiles, and optional transformer-block hooks). We report wall-clock time and peak VRAM for the same step budget and batch size.

### G.4. Ablation: Navi-FlowEdit ($M \equiv 1$) versus Navi-FlowEdit + gate

To isolate the contribution of the optional gate from scale-density allocation, we fix the same effective window $\mathcal{U}_{\mathrm{eff}}$, the same decoupled schedule family, match step budget and random seeds, and disable online navigation. The corresponding matched rows are reported directly in Table 1, so we do not duplicate a separate table here.

*Table 6.* Results on ImgEdit-Bench across compatible editors. Navi-X denotes applying the same decoupled controller to editor X; both Navi rows here use $M \equiv 1$.

| Method | Type | Add | Remove | Replace | Adjust | Background | Style | Action | Extract | Compose | Basic Avg.↑ | UGE Avg.↑ |
|---|---|---|---|---|---|---|---|---|---|---|---|---|
| InfEdit | Diffusion-based training-free | 2.33 | 1.79 | 2.21 | 2.05 | 2.31 | 2.29 | 1.61 | 3.96 | 2.30 | 2.43 | 2.96 |
| **Navi-InfEdit** ($M \equiv 1$) | Diffusion-based training-free | **2.42** | **1.92** | **2.40** | **2.07** | **2.55** | **2.31** | **2.12** | **4.10** | **2.51** | **2.57** | **3.02** |
| FlowAlign | Flow-based training-free | 4.14 | **2.41** | 2.62 | **2.73** | 2.74 | **2.63** | 2.42 | **3.91** | 2.26 | 3.01 | 3.04 |
| **Navi-FlowAlign** ($M \equiv 1$) | Flow-based training-free | **4.28** | 2.37 | **2.80** | 2.63 | **2.99** | 2.39 | **2.81** | 3.84 | **2.26** | **3.04** | **3.11** |

The optional gate substantially improves preservation, while in this setting it does not reduce CLIP compliance; the semantic scores are slightly higher, although the primary benefit is preservation.

We emphasize that $\mathcal{U}_{\text{eff}}$ is operationalized as a fixed tail window and is not selected per instance from $\rho(u)$. Therefore, masking affects the local update direction via Eq. (45) and the diagnostic statistics (e.g., $\rho_k$), but it does not change the visited scale window in our reported experiments.

# H. ImgEdit-Bench Results Across Compatible Editors

PIE-Bench isolates preservation under source-region masks, but it does not cover instruction-style, UGE, or multi-turn editing. We therefore additionally evaluate on ImgEdit-Bench (Ye et al., 2025). In our infrastructure, this benchmark includes the nine Basic single-turn categories (add, remove, replace, adjust, background, style, action, extract, compose), the official UGE suite, and multi-turn subsets for content memory, content understanding, and version backtracking.

Prompt-pair differential editors require both a source description and a target description, whereas ImgEdit-Bench natively provides a source image and an edit instruction. To adapt the benchmark fairly, we use a single processed-annotation layer generated with Qwen2.5-VL (Bai et al., 2025), which converts each sample into a shared (`src_prompt, tar_prompt`) pair. InfEdit, FlowAlign, and their Navi variants all consume this same prompt-pair annotation. All reported scores follow the official ImgEdit-Judge protocol; we report per-category Basic scores, the Basic average, and the UGE average. The multi-turn subset is retained as review manifests for qualitative inspection rather than collapsed into a single scalar.

Table 6 shows positive average gains for both compatible editors. Both Navi rows are ungated, so this evidence is independent of the optional gate used in the default FlowEdit system. For InfEdit, the gains are category-wide in this table; for FlowAlign, the gains are average-level rather than universal across categories, with the clearest positive changes in background, action, and replace. NaviEdit is therefore not a claim of per-category dominance, but a portable inference-time control principle that can improve compatible editors on average.

# I. Failure Cases and Scope

Navi improves *how* the trajectory allocates computation, but it does not by itself solve all support-estimation or scene-reasoning failures. We repeatedly observe two representative failure modes.

**Incomplete semantic migration.**   When the editable support is estimated too conservatively, or when the effective window is too preservation-biased for a difficult replacement, the result can remain semantically intermediate: some target attributes appear, but source morphology or texture is still partially retained. This failure mode is most visible in fine-grained replacements and is especially relevant to gated variants such as *Navi-FlowEdit + gate*.

**Relational inconsistency in complex scenes.**   Navi does not impose explicit scene-graph, geometric, or reflection consistency. In scenes involving mirrors, reflections, repeated objects, or strong inter-object relations, a local edit can be plausible in isolation while remaining globally inconsistent with the rest of the scene. This limitation persists even when the controller successfully avoids global drift.

These failures clarify the scope of our contribution. Navi addresses trajectory control under a fixed budget; it is not a full semantic parser or a world-modeling module. Future work should combine the decoupled controller with stronger support estimation, instance-adaptive windowing, and explicit relational reasoning.

## J. Navi controller inference pipeline

Algorithm 1 outlines the complete inference procedure for prompt-pair differential editors. Unlike standard diffusion sampling, the Navi controller explicitly decouples the editing progress (denoted by $p$) from the model's scale schedule (denoted by $u$). In each iteration, the algorithm enforces the *self-consistency contract* by strictly using the same scale coordinate $u_k$ for constructing anchors, querying the velocity field, and defining the update step size. This ensures that the integration remains stable even when the navigation density is dynamically adjusted within the effective window.

---

**Algorithm 1** Navi controller inference (strict model-call budget, self-consistent steps)

---

**Require:** Source latent $x_{\mathrm{src}}$; conditions $(c_{\mathrm{src}}, c_{\mathrm{tar}})$; velocity field $v_\theta(z, t, c)$; edit steps $K$; optional online navigation; optional internal gate (no external masks).

**Ensure:** Edited latent $x_K$.

Retrieve scheduler timesteps $\{t^{(n)}\}_{n=0}^{N-1}$; compute normalized scales $\{u^{(n)}\} \subset [0, 1]$ and sort to obtain a monotone path $(t_{\mathrm{path}}[n], u_{\mathrm{path}}[n])$ from noisy→clean.

Define linear interpolation $(t, u) \leftarrow \mathrm{Interp}(t_{\mathrm{path}}, u_{\mathrm{path}}; p)$ for $p \in [0, 1]$.

Initialize $x_0 \leftarrow x_{\mathrm{src}}$, $p_0$ (tail start). Set $\Delta V_{-1} \leftarrow$ NONE.

**for** $k = 0, \dots, K - 1$ **do**

  $(t_k, u_k) \leftarrow \mathrm{Interp}(\cdot; p_k)$ and sample fresh $\epsilon_k \sim \mathcal{N}(0, I)$.

  $z_k^{\mathrm{src}} \leftarrow (1 - u_k)x_{\mathrm{src}} + u_k\epsilon_k, \quad z_k^{\mathrm{tar}} \leftarrow x_k + (z_k^{\mathrm{src}} - x_{\mathrm{src}})$.

  Query once with a concatenated batch $\{z_k^{\mathrm{src}}, z_k^{\mathrm{tar}}\}$ at the same time $t_k$ to get $V_k^{\mathrm{src}} \leftarrow v_\theta(z_k^{\mathrm{src}}, t_k, c_{\mathrm{src}})$ and $V_k^{\mathrm{tar}} \leftarrow v_\theta(z_k^{\mathrm{tar}}, t_k, c_{\mathrm{tar}})$.

  $\Delta V_k \leftarrow V_k^{\mathrm{tar}} - V_k^{\mathrm{src}}$.

  Optional internal gate (no extra queries): compute $M_k$ from model activations at this step; default $M_k \equiv 1$.

  $\Delta V_k^{\mathrm{eff}} \leftarrow M_k \odot \Delta V_k$.

  $\Delta p_{\min} \leftarrow (1 - p_k)/(K - k)$.

  **if** online navigation enabled **then**

    Compute risk from already-available signals, e.g., $\omega_k \leftarrow 0$ if $\Delta V_{k-1} =$ NONE else $1 - \cos(\Delta V_k^{\mathrm{eff}}, \Delta V_{k-1}^{\mathrm{eff}})$, and a magnitude term from $\|\Delta V_k^{\mathrm{eff}}\|$.

    $\Delta p_k \leftarrow \Delta p_{\min} \cdot \mathrm{Speed(risk)}$.

  **else**

    $\Delta p_k \leftarrow \Delta p_{\min}$.

  **end if**

  **if** $k < K - 1$ **then**

    $p_{k+1} \leftarrow \min\{1 - \epsilon_p, \ p_k + \Delta p_k\}$.

  **else**

    $p_{k+1} \leftarrow 1$.

  **end if**

  $p_{k+1} \leftarrow \max\{p_k, \min\{1, p_{k+1}\}\}$.

  $(t_{k+1}, u_{k+1}) \leftarrow \mathrm{Interp}(\cdot; p_{k+1})$.

  $\Delta u_k \leftarrow u_{k+1} - u_k, \quad x_{k+1} \leftarrow x_k + \Delta u_k \Delta V_k^{\mathrm{eff}}$.

**end for**

return $x_K$

---

## K. Societal Impacts

The development of high-fidelity, training-free editing frameworks like NaviEdit presents significant opportunities while also necessitating a discussion of societal and ethical considerations.

**Positive Applications and Broader Impacts.** Our primary motivation for developing NaviEdit is to enhance the controllability and accessibility of high-end creative tools. The method's core advantages—its training-free nature, compatibility with modern flow-based models (Esser et al., 2024), and ability to operate efficiently under a fixed compute budget—make powerful generative editing accessible to a broader audience without the need for expensive model fine-tuning or heavy inversion branches. We envision our work empowering artists, designers, and content creators by providing a reliable tool

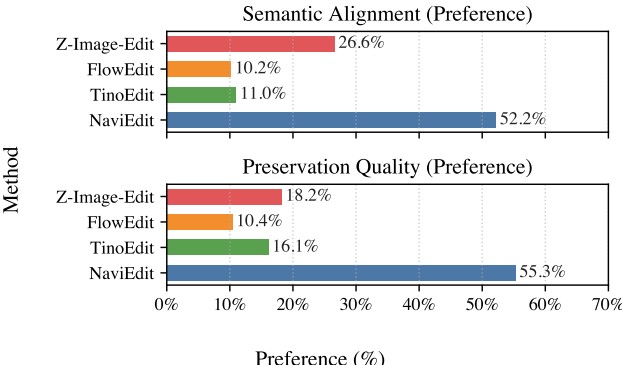

*Figure 15.* **User Study Results.** We conducted a blind preference study with 150 participants. *Navi-FlowEdit + gate* is preferred over Z-Image-Edit, FlowEdit, and Tino-Edit in both Semantic Alignment (52.2%) and Preservation Quality (55.3%).

for rapid prototyping and concept visualization. For instance, a designer could instantly visualize drastic semantic alterations (e.g., changing a material or object class) while trusting that the global layout and non-edited regions will remain structurally intact. This reduces the technical barrier to complex image manipulation, fostering greater creative expression.

**Ethical Considerations and Potential for Misuse.** Like all high-fidelity generative models, NaviEdit carries the risk of misuse. As demonstrated in our experiments, the ability to execute intense semantic changes while maintaining high structural fidelity could be exploited to generate deceptive content or spread misinformation. The high quality of background preservation achieved by our method might make such forgeries more convincing, as the unaltered context adds to the perceived authenticity of the manipulated content.

Furthermore, NaviEdit is a training-free method that operates directly on pre-trained text-to-image models (e.g., Stable Diffusion 3). It does not, by itself, correct any inherent societal biases (e.g., related to race, gender, or culture) that may be present in these foundational models. As such, edits performed by NaviEdit may reflect or even amplify these underlying biases, depending on the prompts and the specific model employed.

**Mitigation and Author Statement.** We strongly condemn the use of our technology for any deceptive or harmful purpose. Our work is intended for creative and assistive applications, aimed at augmenting human creativity, not replacing it or enabling deception. We believe the best mitigation strategy lies in the concurrent development of robust detection tools for synthetic media, as well as in fostering public awareness and critical media literacy. We encourage the research community to continue to prioritize the development of ethical guidelines and safeguards alongside the advancement of generative capabilities.

## L. User Study

To complement our quantitative analyses on PIE-Bench, we conducted a formal user study to assess human perceptual preference. An example of the evaluation form shown to participants is provided in Figure 16. Automated metrics often struggle to capture the holistic "quality" or "naturalness" of an edit. This study therefore tests whether the deployed system in the main paper, *Navi-FlowEdit + gate*, is preferred by human observers over its main comparison methods.

For the study setup, we recruited 150 participants with diverse backgrounds. We presented them with a four-way blind comparison, showing the original image, a text prompt, and four edited results from *Navi-FlowEdit + gate* (ours), Z-Image-Edit (Cai et al., 2025), FlowEdit (Kulikov et al., 2025), and Tino-Edit (Chen et al., 2024). The order of all results was randomized to prevent bias.

Participants rated each edited result on a 1–5 rating scale. The two primary criteria used for the reported analysis were Semantic Alignment, which measures how well the edited image matches the target prompt, and Preservation Quality, which measures naturalness, background preservation, and the absence of artifacts in non-edited regions.

For each participant, prompt, and reported criterion, we identified the method or methods receiving the highest rating. If a single method received the highest rating, it received one preference unit. If multiple methods tied for the highest rating, the

unit was split equally among the tied methods. This yields 4,500 total preference units (150 participants × 30 prompts) for each reported criterion.

For **Semantic Alignment**, *Navi-FlowEdit + gate* was the clear winner, preferred in **52.2%** of comparisons. This outperformed Z-Image-Edit (26.6%), while Tino-Edit (11.0%) and FlowEdit (10.2%) lagged considerably. This indicates that the decoupled rollout is perceived as executing the prompt more faithfully than the compared methods.

For **Preservation Quality**, *Navi-FlowEdit + gate* also achieved the top position, securing **55.3%** of the vote. This indicates that users perceived it as more stable and artifact-free than Z-Image-Edit (18.2%) and Tino-Edit (16.1%). FlowEdit (10.4%) was frequently penalized for background distortion or drift.

Overall, the user study is consistent with the main quantitative results. *Navi-FlowEdit + gate* is the only method that ranks first in both categories, indicating that the improved trade-off is visible to human observers rather than only to automated metrics.

## M. More Qualitative Results

We provide additional qualitative comparisons in Figure 17. These examples further support the quantitative findings presented in the main paper.

In these qualitative examples, *Navi-FlowEdit + gate* produces high-fidelity results that follow the target prompt while maintaining strong background preservation. This contrasts with coupled schedules such as *FlowEdit*, which more often introduce drift or background artifacts when asked to execute large semantic changes. The qualitative results are consistent with the central mechanism of the paper: decoupled navigation keeps the rollout inside an effective scale window where the generative prior is more stable.



# User Study Evaluation Form

**Purpose:** This study aims to assess the effectiveness and perceptual quality of various image editing methods. Your feedback will help improve the quality of diffusion-model-based editing techniques.

**Instructions:**

1) You will see an original image followed by several edited versions generated by different methods.

2) Please rate each processed image based on the criteria provided below.

3) Use the scale from 1 (Poor) to 5 (Excellent) for your rating.

**Evaluation Criteria:**

1) **Target Semantics Alignment:** Assessing how accurately and convincingly the edited image reflects the intended semantic change (e.g., "turn a dog into a cat," "add sunglasses").

2) **Background Preservation:** Evaluating how well the non-edited regions of the image remain consistent with the original (e.g., no unwanted distortions, color shifts, or texture changes in the background).

3) **Overall Coherence:** Assessing the visual plausibility and naturalness of the entire edited image, considering both the edit and the background.

**Image Evaluation Example:** (Below is a demonstration of one group of images to serve as an example for the evaluation process.)

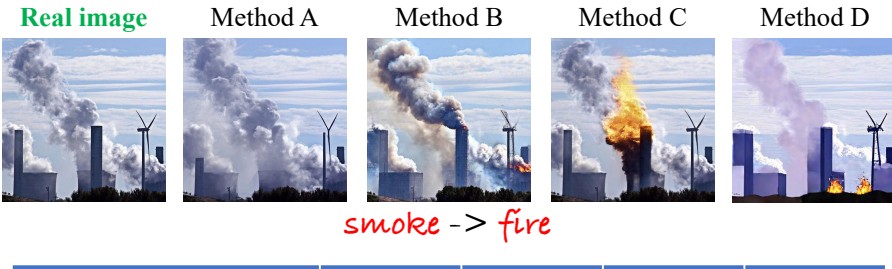

| Criteria | Method A | Method B | Method C | Method D |
|---|---|---|---|---|
| **Semantics Alignment** | | | | |
| **Background Preservation** | | | | |
| **Overall Coherence** | | | | |

Groups 2 to 30 have been omitted in this section for brevity.



*Figure 16.* **User Study Form.**

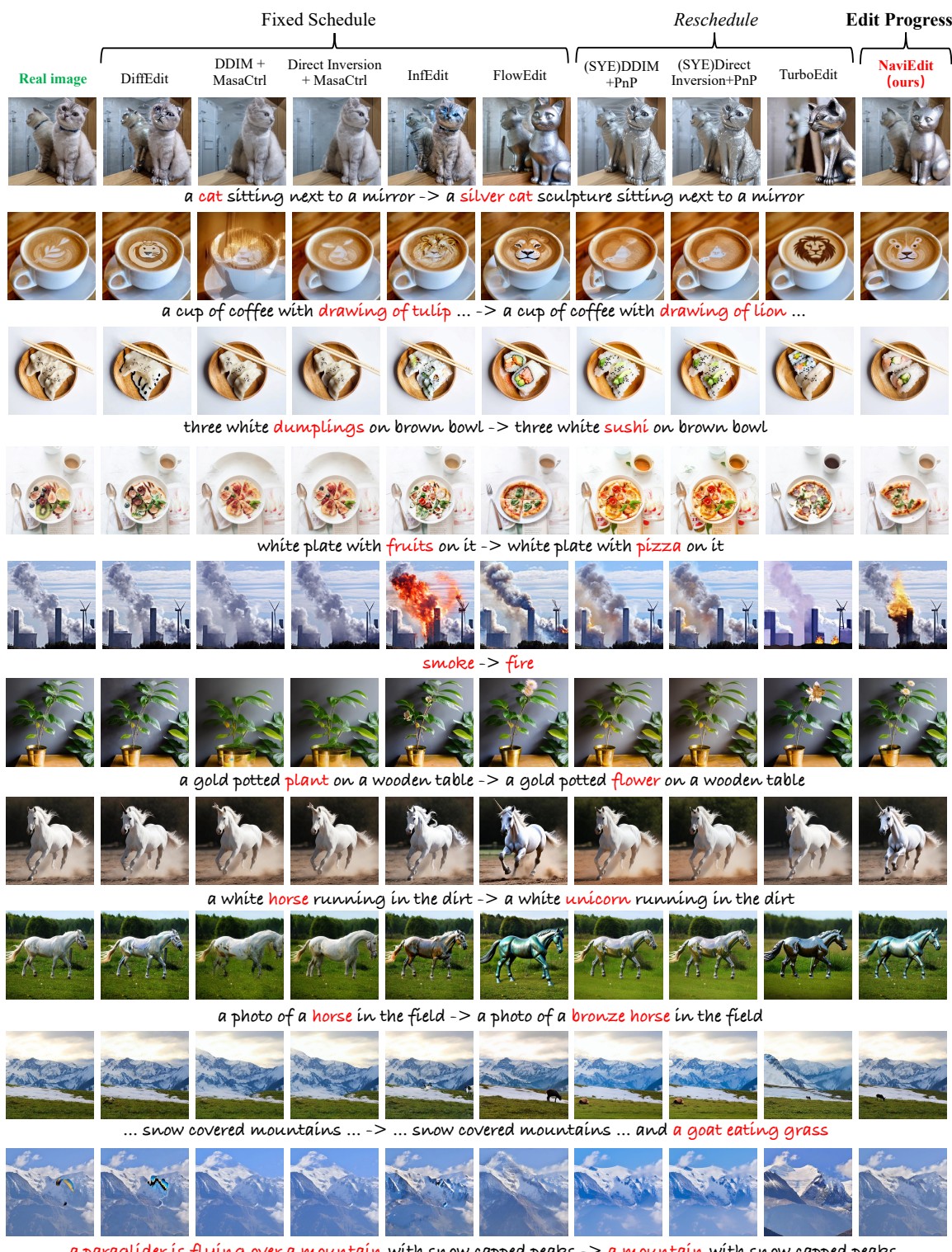

*Figure 17.* **Additional Qualitative Comparisons.** These examples illustrate that *Navi-FlowEdit + gate* can execute semantic edits while preserving non-edited regions, whereas the compared baselines may exhibit background leakage or structural degradation.

