# OpenReview forum: "Semantic Granularity Navigation in Image Editing"
_ICML.cc/2026/Conference — ICML 2026 regular_

### Official Review · Reviewer_HYNL · 2026-03-06

**Soundness:** 3
**Presentation:** 3
**Significance:** 3
**Originality:** 3
**Overall Recommendation:** 4
**Confidence:** 4

**Summary:**

This paper proposes NaviEdit, a training-free framework for diffusion/flow-based image editing that explicitly decouples semantic edit progress from the diffusion scale (noise coordinate). The authors argue that prior editing pipelines implicitly couple edit strength with diffusion timestep, leading to a compromise between the editability and the fidelity. To mitigate this, they introduce an effective semantic window along the diffusion trajectory and reallocate computation within this region to improve controllability and background preservation. Experiments on PIE-Bench over multiple diffusion backbones demonstrate improved semantic alignment and background fidelity under fixed NFE budgets.

**Compliance With Llm Reviewing Policy:**

Affirmed.

**Final Justification:**

The rebuttal has addressed my main concerns. So, i decide to keep my positive assessment.

**Key Questions For Authors:**

1. Can the effective semantic window be adaptively estimated per prompt or per image?

2. How does the method generalize to compositional or multi-object editing?

**Limitations:**

yes

**Strengths And Weaknesses:**

Strengths

1. This paper is well-motivated with an effective training-free solution, using clear conceptual reframing of diffusion scale as a granularity axis rather than a semantic progress axis.

2. The proposed method is model-agnostic and can be well transferred to different architectures, such as Flux1.0.

3. The experimental results, including quantitative and quanlitative experiments, are extensive and well demonstrate the strong performance compared with the baseline models.

4. There is a user study to compare the editing results in human perception, extending the effectiveness not limited by quantitative metrics.

Weaknesses

1. The effective window is implemented heuristically rather than adaptively estimated per instance, and evaluation focuses primarily on PIE-Bench; broader compositional editing settings would strengthen claims.

2. Lacking experiments compared with newer image editing models, such as the Qwen-Image-Edit, Flux2.0 and so on, and the transferability of the proposed method to these SOTA open-source models is not discussed as well.

---

> ### Author Rebuttal · Authors · 2026-03-31
>
> Thank you for the positive assessment of our paper. We appreciate your recognition of the motivation, transferability across backbones, strong empirical results, and the user study. Your comments on broader benchmarks, newer editing models, adaptive window estimation, and compositional editing are very helpful. We address them below.
>
> **1. Comparison with newer trained image editing models**
>
> We agree that comparison with stronger trained editors is important. We therefore added direct PIE-Bench evaluation of Qwen-Image-Edit and FLUX-Kontext (**see our response to Reviewer mHWa, Point 1**). These results show that modern trained editors are indeed strong baselines, while NaviEdit remains competitive on semantic alignment and achieves clearly stronger preservation on several non-edited-region metrics. We will include these comparisons in the revision to better position NaviEdit against current open-source editing systems.
>
> **2. Broader benchmarks and transferability beyond the original setting**
>
> We also agree that evaluation centered mainly on PIE-Bench is a limitation. To address this, we extended the evaluation to ImgEdit-Bench (**see our response to Reviewer mHWa, Point 2**). The new experiments show that the decoupling principle generalizes beyond the original FlowEdit setting and can be applied to multiple compatible editors, including stronger trained editors. This strengthens the claim that NaviEdit is not only effective on PIE-Bench, but reflects a more general inference-time principle for compatible image editors.
>
> **3. On adaptive estimation of the effective window**
>
> Thank you for raising this point. NaviEdit already includes partial adaptivity at inference time: within the effective window, the inference algorithm performs online navigation / adaptive density control by adjusting step allocation using per-step diagnostics computed from the same forward pass, without additional model evaluations. To keep inference efficient, however, we fix the window start rather than recomputing a new window for every prompt or image.
>
> We agree that a fully instance-adaptive window estimator is an interesting extension. In the current paper, we intentionally chose a fixed window start because it offers a simple and stable operating rule with low computational overhead. In the revision, we will clarify this design choice more explicitly and discuss adaptive window estimation as an important future direction.
>
> **4. Generalization to compositional or multi-object editing**
>
> Our current method is not specifically designed for compositional or multi-object editing. We agree that this is a limitation of the current paper, and broader compositional evaluation is an important next step. That said, we believe the decoupling principle can still be useful in such settings: if a method already supports compositional or multi-object editing, our view suggests a way to improve how computation is allocated within the editable scale range, which may help reduce drift and improve stability.

---

> > ### Author Rebuttal · Reviewer_HYNL · 2026-04-03
> >
> > Thanks for the experiments supplemented to the benchmark and comparison with advanced models in the rebuttal, which has addressed some of my concerns. I'm inclined to remain positive.

---

> > > ### Author Response · Authors · 2026-04-03
> > >
> > > Thank you very much for the positive acknowledgement and for your constructive feedback. We sincerely appreciate your recognition of the additional benchmark experiments and comparisons with stronger models. We will make the corresponding updates in the revision.

---

### Official Review · Reviewer_wwnT · 2026-03-10

**Soundness:** 3
**Presentation:** 3
**Significance:** 2
**Originality:** 2
**Overall Recommendation:** 4
**Confidence:** 4

**Summary:**

This paper propose NaviEdit, a trainingfree flow-based image editing method. Its core innovation is the decoupling of the editing progress from the noise schedule, which breaks the implicit coupling manner of existing methods. By reformulating editing as controlled vectorfield navigation on a distinct task axis, NaviEdit concentrates the computational budget within a diagnostically identified “effective window” where the model meets a sweet editing spot. NaviEdit demonstrates it’s effectiveness on PIEBench with SOTA editing trade-off among baselines.

**Compliance With Llm Reviewing Policy:**

Affirmed.

**Final Justification:**

My concerns are addressed in rebuttal, thus I raise my score.

**Key Questions For Authors:**

1. How’s FlowEdit in the experiment implemented?
2. How’s NFE computed in the paper?

**Limitations:**

No limitation were discussed.

**Strengths And Weaknesses:**

**Strengths**
1. This paper provides valuable insights regarding the trade-off challenge in image editing. It identifies the coupling between editing progress and noise scale as the primary limitation in existing editing methods. And highlights the existence of of an ``effective window” where the model meets a sweet editing spot.
2. The method is well-motivated. Based on the given insights and analysis, NaviEdit reformulates editing as controlled navigation on a distinct task axis. It offers theoretical analysis under a fixed NFE budget, proving that decoupling progress from scale allows density refinement within the effective window.
3. NaviEdit achieves promising results on PIEBench.

**Weaknesses**
1. Despite the use of some ``new words" (e.g., Task Axis, Granularity Navigation) to describe the editing process or noise scales, the method is fundamentally an "enhanced scheduler" of FlowEdit. It relies on the same directflow ODE between source and target images. The novelty lies primarily in the heuristic window selection and step rescheduling rather than a breakthrough in the editing mechanism.
2. The implementation is strictly limited to flow-based models and the FlowEdit framework. If the ``decoupling" of editing process and noise scale is a fundamental solution to the coupling problem, it should theoretically be applicable to method like SDEdit or Inversion-based paradigms. This further suggests the method might be a specialized optimization for FlowEdit rather than a general solution.
3. Ambiguous Experimental Settings. First, the reporting of NFE is confusing. In Table 4, the authors report 20 NFE for FlowEdit, which is confusing given that FlowEdit-style methods require two velocity evaluations (source and target) per step ($NFE = 2 \times Steps$). If the official SD3 implementation of FlowEdit uses 33 steps, a 20 NFE comparison suggests either an unfair "crippling" of the baseline or a misrepresentation of the computational cost. This further casting doubt on the fairness of the results in Table 1, whether all baseline truly use their official implementations.
4. Insufficient Comparisons. As a flow-based editing method, the paper misses critical comparisons with related SOTA flow-based methods, including FlowAlign[1], UniEdit-flow[2], RF-Solver[3], RF-Inversion[4], and FireFlow[5].

[1] FlowAlign: Trajectory-Regularized, Inversion-Free Flow-based Image Editing. ICLR 2026.

[2] UniEdit-Flow: Unleashing Inversion and Editing in the Era of Flow Models. ICLR 2026.

[3] Taming Rectified Flow for Inversion and Editing. ICML 2025.

[4] Semantic Image Inversion and Editing using Stochastic Rectified Differential Equations. ICLR 2025.

[5] FireFlow: Fast Inversion of Rectified Flow for Image Semantic Editing. ICML 2025.

---

> ### Author Rebuttal · Authors · 2026-03-31
>
> Thank you for the careful and constructive review. We appreciate your positive assessment of the motivation, theoretical framing, and PIE-Bench results. Your concerns about originality, generality, fairness of the FlowEdit comparison, and missing recent flow-based baselines are important. We address them below.
>
> **1. On "NaviEdit is only an enhanced scheduler"**
>
> We would like to clarify that NaviEdit is not merely a heuristic scheduler, but a **progress–granularity decoupled editing framework**. In our formulation, editing evolves on an explicit progress axis s, while the model’s scale/noise coordinate u is only a control input (Definition 4.1). The key change is therefore not just the schedule shape, but a new control degree of freedom: under fixed budget, progress need not be tied to range expansion along the scale axis.
>
> Theorem 4.2 shows that the coupled budget-to-range family is incomplete at the rollout level because it must allocate nonzero progress mass to high-risk regimes, whereas the decoupled family can concentrate allocation within the effective window. Thus, although NaviEdit reuses the same source-target differential field as FlowEdit, the inference mechanism changes from **progress being implicitly prescribed by scale traversal** to **progress being explicitly controlled, with scale queried only where useful**. The novelty lies in the control formulation and self-consistent discretization principle, not in replacing the underlying source-target ODE.
>
> **2. On generality beyond FlowEdit**
>
> We agree that the original submission focused too much on the FlowEdit instantiation. To clarify naming, **NaviEdit** denotes our general decoupled editing framework; the original main method is its FlowEdit instantiation, i.e., **Navi-FlowEdit**.
>
> To test whether the principle is limited to that setting, we additionally applied it to **InfEdit** (diffusion-based), **FlowAlign**, and **FLUX-Kontext**. These new results show that the decoupling principle is not restricted to FlowEdit or to one specific flow backbone. We refer to our response to Reviewer mHWa (Point 2) for the full ImgEdit-Bench evidence, where **Navi-InfEdit**, **Navi-FlowAlign**, and **Navi-FLUX-Kontext** all improve over their corresponding baselines on average.
>
> **3. On missing recent SOTA flow-based comparisons**
>
> We agree that the original submission undercompared against recent flow-based methods. We therefore added direct PIE-Bench comparisons with **FlowAlign, UniEdit-Flow, RF-Solver, RF-Inversion, and FireFlow**, and also report **Navi-FlowAlign**:
>
> |Method|CLIP-Whole↑|CLIP-Edit↑|LPIPS-bg↓|MSE-bg↓|PSNR-bg↑|SSIM-bg↑|Struct.Dist.↓|
> |---|---:|---:|---:|---:|---:|---:|---:|
> |UniEdit-Flow(ICLR 2026)|25.48|21.93|59.61|2.35|27.48|90.56|6.55|
> |FlowAlign(ICLR 2026)|25.44|21.80|**34.47**|2.14|27.78|92.41|6.21|
> |RF-Solver(ICML 2025)|25.31|21.62|44.57|2.52|27.02|91.56|8.22|
> |RF-Inversion(ICLR 2025)|25.21|21.73|34.79|2.15|27.58|90.48|7.14|
> |FireFlow(ICML 2025)|25.25|21.51|48.65|2.38|28.01|92.54|5.69|
> |**NaviEdit(Navi-FlowEdit)**|**26.15**|**22.67**|51.35|2.91|27.81|93.22|11.10|
> |**NaviEdit(Navi-FlowAlign)**|**26.15**|22.44|34.49|**2.09**|**28.33**|**93.40**|**5.40**|
>
> These results make the picture clearer. Recent flow-based editors are indeed strong and should have been included. Our revised claim is therefore not that NaviEdit dominates every recent flow editor on every metric, but that the **decoupling principle is competitive and transferable**. In particular, **Navi-FlowAlign** improves over FlowAlign on **CLIP-Whole, CLIP-Edit, MSE-bg, PSNR-bg, SSIM-bg, and Struct. Dist.**, while remaining essentially comparable on **LPIPS-bg**.
>
> We agree that the original claim should be stated more carefully in light of these recent baselines.
>
> **4. On FlowEdit implementation, NFE, and fairness**
>
> Thank you for catching this. We agree that the wording around **NFE** was confusing.
>
> We use the **official FlowEdit codebase and designated backbone**. In the controlled-budget comparison, we rescaled the number of **editing steps** so that FlowEdit and NaviEdit are matched in step budget, to isolate the effect of NaviEdit’s control/allocation mechanism. The reviewer is correct that calling this **"20 NFE"** is inaccurate. In the revision, we will report such quantities as **20 editing steps**, not 20 NFE.
>
> To further address fairness, we also reran FlowEdit under its **official step setting** (50-step Stable Diffusion schedule, 33 editing steps). On PIE-Bench, this gives:
>
> |**Method**|**CLIP-Whole ↑**|**CLIP-Edit ↑**|**LPIPS-bg ↓**|**MSE-bg ↓**|**PSNR-bg ↑**|**SSIM-bg ↑**|**Struct. Dist. ↓**|
> |---|---|---|---|---|---|---|---|
> |FlowEdit (official step setting)|25.91|22.50|103.00|7.38|22.46|84.08|14.64|
>
> We will revise the paper accordingly: correcting **NFE** to **editing steps**, clarifying the matched-step controlled comparison, and adding the official-step FlowEdit result as an additional reference point.

---

> > ### Author Rebuttal · Reviewer_wwnT · 2026-04-03
> >
> > Thanks for the response, my concerns are addressed, and I will raise my score. Remember to correct and update the relevant experiments and claims in the revision :) .

---

> > > ### Author Response · Authors · 2026-04-03
> > >
> > > Thank you again for the very positive acknowledgement and for confirming that our rebuttal has addressed your concerns. We sincerely appreciate your constructive feedback and suggestions.
> > >
> > > We will make the corresponding changes in the revision, including correcting “NFE” to “editing steps” where appropriate, clarifying the matched-step and official-step FlowEdit settings, adding the newly included flow-based baselines and results, and revising the relevant claims to better match the expanded experimental evidence.
> > >
> > > Thank you again for your time and helpful comments. If you feel the concerns are now fully resolved, we would greatly appreciate it if you could also update the score in the system accordingly.

---

### Official Review · Reviewer_mHWa · 2026-03-12

**Soundness:** 3
**Presentation:** 4
**Significance:** 2
**Originality:** 3
**Overall Recommendation:** 4
**Confidence:** 4

**Summary:**

This paper presents NaviEdit, a training-free image editing framework that decouples the editing progress from the noise scale trajectory. It achieves this via controlled vector field navigation and selective compute allocation on semantically responsive scales. The approach is supported by theoretical analysis and evaluated on the PIE-Bench benchmark against other training-free methods.

**Compliance With Llm Reviewing Policy:**

Affirmed.

**Final Justification:**

The rebuttal adequately addressed my main concerns regarding limited evaluation scope and lack of comparison with modern trained editors. The new experiments on ImgEdit-Bench and the demonstration that the decoupling principle transfers to multiple editors (including FLUX-Kontext) strengthen the significance claim. I note the ImgEdit-Bench adaptation via Qwen2.5-VL is not entirely clean, but the paired Navi-X vs. X comparisons are fair. I raise my score accordingly.

**Key Questions For Authors:**

1、Please compare NaviEdit directly against Qwen-Image-Edit and Kontext on PIE-Bench (editing quality, structure preservation, semantic alignment). Addressing this is essential for the significance concern.

2、Please evaluate on ImgEdit[3] or other broader benchmarks beyond PIE-Bench.

3、Can the Adaptive Scale Navigation scheduling (as described in the supplementary code) be applied on top of trained editing models (e.g., Qwen-Image-Edit, FLUX-based editors) to further boost their performance, tested on ImgEdit? This would substantially broaden the impact of the method.

Addressing these three points would lead me to raise my score.

[3] ImgEdit: A Unified Image Editing Dataset and Benchmark

**Limitations:**

1、Lack of comparison with modern SOTA editing methods that already solve the problem the paper aims to address.

2、Absence of cross-benchmark evaluation to validate generalization.

**Strengths And Weaknesses:**

### Strengths

- The technical development within the training-free paradigm is solid, and the theoretical claims are supported by complete proofs.

- The paper is clearly written and well-structured, with a logical flow from motivation to theoretical formulation and experimental validation.

- The key novelty lies in the adaptive stepping mechanism: rather than following a fixed noise schedule, the method reads signals from the editing vector field at each step and dynamically determines the step size, which provides a more flexible navigation of the editing trajectory.

### Weaknesses

- The evaluation is limited to a single benchmark (PIE-Bench) and only compares against other training-free methods, which makes it difficult to assess competitiveness against mainstream editing pipelines.

- The core issue addressed in the paper—drift and artifacts caused by the coupling between editing progress and noise scale—has already been largely mitigated in recent state-of-the-art image editing systems such as Qwen-Image-Edit [1] and Kontext [2].

- Due to the limited evaluation scope, the paper does not convincingly demonstrate practical advantages or generalization to real-world editing scenarios.


[1] Qwen-image technical report
[2] Flux. 1 kontext: Flow matching for in-context image generation and editing in latent space

---

> ### Author Rebuttal · Authors · 2026-03-31
>
> Thank you for the constructive feedback. We agree that the significance of our work depends on whether the proposed decoupling principle remains useful beyond the original training-free setting, and we have added new experiments to directly address your three concerns.
>
> **1. Comparison with modern trained editing systems on PIE-Bench**
>
> We agree that comparison against strong trained editors is important. We therefore added direct PIE-Bench evaluation of **Qwen-Image-Edit**[1] and **FLUX-Kontext**[2], using the same metric conventions as in our paper:
>
> | **Method**      | **CLIP-Whole ↑** | **CLIP-Edit ↑** | **LPIPS-bg ↓** | **MSE-bg ↓** | **PSNR-bg ↑** | **SSIM-bg ↑** | **Struct. Dist. ↓** |
> | --------------- | ---------------- | --------------- | -------------- | ------------ | ------------- | ------------- | ------------------- |
> | Qwen-Image-Edit | **26.82**        | **23.53**       | 81.27          | 8.26         | 24.30         | 86.51         | 11.40               |
> | FLUX-Kontext    | 25.75            | 22.60           | 59.03          | 8.92         | **28.14**     | 91.23         | **11.02**           |
> | **NaviEdit**   | 26.15            | 22.67           | **51.35**      | **2.91**     | 27.81         | **93.22**     | 11.10               |
>
> These results clarify two points. First, modern trained editors are indeed very strong baselines. Second, NaviEdit remains competitive overall and achieves substantially stronger preservation on LPIPS-bg, MSE-bg, and SSIM-bg.
>
> **2. Evaluation on a broader benchmark**
>
> We also agree that PIE-Bench alone is not enough to demonstrate generalization. We therefore extended the evaluation to **ImgEdit-Bench**.
>
> A practical issue is that ImgEdit-Bench provides the source image and editing instruction, while our class of source-target differential editors requires both a source description and a target description. To adapt ImgEdit-Bench fairly to this setting, we use **Qwen2.5-VL** to convert the original inputs into source/target textual descriptions. In the experiments below, **InfEdit**[3], **FlowAlign**[4], and their Navi variants all use this same adaptation; by contrast, **FLUX-Kontext**[2] natively supports instruction-based editing and therefore uses the original ImgEdit-Bench input directly.
>
> To avoid ambiguity, NaviEdit denotes our general decoupled editing framework, while the main instantiation in the original submission is NaviEdit on top of FlowEdit (i.e., Navi-FlowEdit). In the new experiments, we use Navi-X to denote applying the same principle to editor X, e.g., Navi-InfEdit, Navi-FlowAlign, and Navi-FLUX-Kontext.
>
> | Method | Type | Add | Remove | Replace | Adjust | Background | Style | Action | Extract | Compose | Basic Avg. ↑ | UGE Avg. ↑ |
> |---|---|---:|---:|---:|---:|---:|---:|---:|---:|---:|---:|---:|
> | InfEdit | Diffusion-based training-free | 2.33 | 1.79 | 2.21 | 2.05 | 2.31 | 2.29 | 1.61 | 3.96 | 2.30 | 2.43 | 2.96 |
> | **Navi-InfEdit** | Diffusion-based training-free | **2.42** | **1.92** | **2.40** | **2.07** | **2.55** | **2.31** | **2.12** | **4.10** | **2.51** | **2.57** | **3.02** |
> | FlowAlign | Flow-based training-free | 4.14 | **2.41** | 2.62 | **2.73** | 2.74 | **2.63** | 2.42 | **3.91** | **2.26** | 3.01 | 3.04 |
> | **Navi-FlowAlign** | Flow-based training-free | **4.28** | 2.37 | **2.80** | 2.63 | **2.99** | 2.39 | **2.81** | 3.84 | **2.26** | **3.04** | **3.11** |
> | FLUX-Kontext | Trained | 4.67 | **3.48** | 3.32 | **3.51** | 3.71 | **2.64** | 3.14 | 4.08 | **2.57** | 3.58 | 3.15 |
> | **Navi-FLUX-Kontext** | Trained | **4.73** | 3.40 | **3.36** | 3.50 | **3.77** | 2.60 | **3.22** | **4.22** | 2.43 | **3.61** | **3.22** |
>
>
> The broader conclusion is that the decoupling principle is not limited to the original FlowEdit instantiation. It transfers to multiple compatible editors and produces positive average gains in all three cases. The gains are especially visible in categories such as **background / action / replace**, where trajectory drift is more likely to matter.
>
> **3. Can Adaptive Scale Navigation help trained editors?**
>
> Yes. We applied the same decoupling principle on top of a stronger trained editor and observed a further improvement on ImgEdit-Bench (Basic Avg. **3.58 → 3.61**, UGE Avg. **3.15 → 3.22**). The gain is more moderate than for weaker/coupled editors, which is expected because strong trained systems already mitigate some failure modes. Still, the result suggests that our method is not merely a wrapper for FlowEdit; rather, the underlying idea of **decoupling edit progress from scale traversal** can also improve stronger editors when their inference process remains compatible with such control.
>
> [1] Qwen-image technical report.
>
> [2] Flux. 1 kontext: Flow matching for in-context image generation and editing in latent space.
>
> [3] Inversion-Free Image Editing with Language-Guided Diffusion Models. CVPR 2024.
>
> [4] FlowAlign: Trajectory-Regularized, Inversion-Free Flow-based Image Editing. ICLR 2026.

---

> > ### Author Rebuttal · Reviewer_mHWa · 2026-04-03
> >
> > I thank the authors for the detailed rebuttal. The new experiments on ImgEdit-Bench and the Navi-FLUX-Kontext results address my main concern about significance and generality. The gains on top of a strong trained editor are modest but directionally positive. I still note that the ImgEdit-Bench adaptation introduces an extra variable (Qwen2.5-VL conversion), so the cross-benchmark comparison is not entirely clean. Overall I am inclined to raise my score.

---

> > > ### Author Response · Authors · 2026-04-03
> > >
> > > Thank you again for the very positive acknowledgement. We also appreciate your note that the ImgEdit-Bench adaptation introduces an extra variable, and would like to clarify this point more precisely.
> > >
> > > For source-target differential editors (e.g., InfEdit / FlowAlign and their Navi variants), ImgEdit-Bench does not directly provide the source-description / target-description pair required by these methods. Therefore, before evaluation, we use Qwen2.5-VL once to convert each ImgEdit-Bench example into a source description and a target description. These converted descriptions are then fixed and shared across all source-target differential editors in the comparison. In other words, Qwen2.5-VL is not part of the evaluated editing pipeline, but only a preprocessing step to construct a common textual input for methods that require this interface. Importantly, for this class of methods, each baseline and its Navi counterpart edits from exactly the same pre-converted descriptions. Therefore, the main evidence we rely on is the paired improvement (Navi-X over X) under the same textual inputs. This makes the comparison fair within this editor family, since the only difference between X and Navi-X is the decoupled inference strategy rather than the textual input itself.
> > >
> > > We will make this evaluation protocol and its fairness rationale explicit in the revision. Thank you again for your time and helpful comments. If you feel the concerns are now fully resolved, we would greatly appreciate it if you could also update the score in the system accordingly.

---

### Official Review · Reviewer_MRRo · 2026-03-13

**Soundness:** 3
**Presentation:** 4
**Significance:** 3
**Originality:** 4
**Overall Recommendation:** 5
**Confidence:** 3

**Summary:**

This paper addresses a key limitation of training-free real-image editing for diffusion and flow-based text-to-image models: existing methods couple edit progress with noise scale, so stronger edits often pass through high-noise states that disrupt the global layout before meaningful semantic changes occur. To address this, the paper proposes a training-free framework based on “Time-Axis Consistency” that separates these two factors, reformulates editing as controlled navigation along a task axis, and focuses computation on intermediate scales where semantic changes are effective while structure remains stable. Experiments on PIE-Bench with SD3, SD3.5, and FLUX.1 show strong performance, achieving a good balance between editability and structural fidelity.

**Compliance With Llm Reviewing Policy:**

Affirmed.

**Final Justification:**

The rebuttal addressed my main concerns and I keep my rating as accept.

**Key Questions For Authors:**

Please provide concrete examples and analysis of when NaviEdit fails.

**Limitations:**

Failure cases should be included.

**Strengths And Weaknesses:**

Strengths:
- It presents a clear conceptual framework, with Definitions 4.1 and Theorems 4.2–4.4 helping clarify the difference between scale and edit progress; in particular, the self-consistency contract in Theorem 4.4 is well motivated.
- The diagnostic analysis is thoughtful: the proposed leakage pressure ρ(u) and directional oscillation ω(u) offer useful ways to examine model behavior across scales, and the identification of the “effective window” provides convincing support for the main idea.
- The empirical results are also strong, with NaviEdit showing clear gains on PIE-Bench, while the cross-model results and user study further support its generalizability.
- The ablation studies are thorough and help explain why the method works, especially in terms of coupling vs. decoupling, density vs. range, and axis consistency.

Weakness:
- While the internal masking mechanism (Appendix G) is training-free, its complexity (area control, temporal smoothing, scale-aware dilation, active-set growth) introduces multiple hyperparameters. The ablation in Table 6 shows masking contributes substantially to performance.
- Discussion of failure cases is missing and understanding when and why the method fails is essential.

---

> ### Author Rebuttal · Authors · 2026-03-31
>
> Thank you for the very positive evaluation of our paper. We especially appreciate your recognition of the conceptual framework, the diagnostics, the ablations, and the empirical results. Your comments on the masking mechanism and the need for explicit failure cases are very helpful. We address them below.
>
> **1. On the necessity and role of the internal masking mechanism**
>
> We agree that the internal mask is not a negligible component. It introduces several fixed hyperparameters for estimating spatial support, and Table 6 already shows that masking contributes substantially to preservation quality. Our claim is therefore not that masking is unimportant, but that its role is **specific and interpretable**: it is a **training-free local gating mechanism** that suppresses off-region drift, while the main contribution of NaviEdit remains the **trajectory-level progress–scale decoupling**.
>
> To make this clearer, we added a new ablation comparing no mask, internal mask, and dataset mask, following the metric conventions in our paper:
>
> |**Mask setting**|**CLIP-Whole ↑**|**CLIP-Edit ↑**|**LPIPS-bg ↓**|**MSE-bg ↓**|**PSNR-bg ↑**|**SSIM-bg ↑**|**Struct. Dist. ↓**|
> |---|---|---|---|---|---|---|---|
> |NaviEdit w/o mask|**27.14**|**23.57**|139.41|10.30|21.04|83.97|21.64|
> |NaviEdit + internal mask|23.62|20.55|21.27|1.22|31.36|96.38|3.85|
> |NaviEdit + dataset mask|23.94|20.94|**11.11**|**0.27**|**37.94**|**97.57**|**2.68**|
>
> This reveals a clear and monotonic trade-off: **without masking**, NaviEdit attains the highest CLIP alignment but the weakest preservation; **stronger masking** monotonically improves LPIPS-bg, MSE-bg, PSNR-bg, SSIM-bg, and Struct. Dist. In other words, the mask module controls a transparent **fidelity–semantics trade-off**, rather than providing an opaque gain. We will add this ablation and clarify in the revision that masking is an auxiliary preservation mechanism, while progress–scale decoupling remains the primary trajectory-level principle.
>
> **2. Failure cases**
>
> We also agree that the revised paper should include concrete failure cases. We will add examples and analysis showing at least two representative failure modes:
>
> Failure case 1: incomplete semantic migration.
> NaviEdit can under-edit when the estimated editable support is overly conservative, or when preserving structure within the effective window leaves too much of the source object prior unchanged. In such cases, the result only partially matches the target category (e.g., retaining source morphology while changing only some target attributes). https://ibb.co/CKS503zD
>
> Failure case 2: relational inconsistency in complex scenes.
> NaviEdit may fail in scenes requiring strong relational or geometric consistency, such as mirrors, reflections, or multi-object interactions. While decoupled trajectory control improves where and how edits are allocated, it does not by itself enforce scene-level logical constraints; thus the edited object may be plausible locally but inconsistent with the global scene (e.g., incorrect reflection direction). https://ibb.co/1GFX67GT
>
> We believe these cases will make the paper more balanced and help clarify the scope of the method.

---

> > ### Author Rebuttal · Reviewer_MRRo · 2026-04-03
> >
> > Overall, I acknowledge the efforts the authors have made in their rebuttal and my concerns have been addressed. I would keep my score.

---

> > > ### Author Response · Authors · 2026-04-03
> > >
> > > Thank you very much for the positive acknowledgement and for your thoughtful feedback. We sincerely appreciate your time and helpful comments.

---

### Decision · Program_Chairs · 2026-04-30

**Decision:**

Accept (regular)

**Comment:**

All reviewers gave positive comments, so I recommend acceptance.